# MesoNet allows automated scaling and segmentation of mouse mesoscale cortical maps using machine learning

Dongsheng Xiao [1], Brandon J. Forys [1,2], Matthieu P. Vanni[1,3] & Timothy H. Murphy [1✉]

Understanding the basis of brain function requires knowledge of cortical operations over wide spatial scales and the quantitative analysis of brain activity in well-defined brain regions. Matching an anatomical atlas to brain functional data requires substantial labor and expertise. Here, we developed an automated machine learning-based registration and segmentation approach for quantitative analysis of mouse mesoscale cortical images. A deep learning model identifies nine cortical landmarks using only a single raw fluorescent image. Another fully convolutional network was adapted to delimit brain boundaries. This anatomical alignment approach was extended by adding three functional alignment approaches that use sensory maps or spatial-temporal activity motifs. We present this methodology as MesoNet, a robust and user-friendly analysis pipeline using pre-trained models to segment brain regions as defined in the Allen Mouse Brain Atlas. This Python-based toolbox can also be combined with existing methods to facilitate high-throughput data analysis.

[1] University of British Columbia, Department of Psychiatry, Kinsmen Laboratory of Neurological Research, Detwiller Pavilion, 2255 Wesbrook Mall, Vancouver V6T 1Z3 British Columbia, Canada. [2] Department of Psychology, Djavad Mowafaghian Centre for Brain Health, University of British Columbia, Vancouver, British Columbia, Canada. [3] Université de Montréal, École d'Optométrie, 3744 Jean Brillant H3T 1P1, Montréal, Québec, Canada. ✉email: thmurphy@mail.ubc.ca

The cerebral cortex is an exquisitely patterned structure that is organized into anatomically and functionally distinct areas[1-4]. A full understanding of cortical activity data requires a reliable approach to segment and classify the different regions of interest based on known anatomical and functional structures. Wide-field cortical calcium imaging in mice has become increasingly popular as it allows efficient mapping of cortical activity over large spatio-temporal scales by expressing cell-type-specific genetically encoded calcium indicators (GECI) such as GCaMP6[5-14]. Automatic registration and segmentation of brain imaging data can greatly improve the speed and precision of data analysis and do not require an expert anatomist. This is particularly crucial when using high-throughput neuroimaging approaches, such as automated mesoscale mouse imaging[15-18], where the amount of data generated greatly exceeds the capacity of manual segmentation.

Deep learning techniques have been used as powerful tools for medical image analysis, including a wide range of applications such as image classification and segmentation[19,20]. For instance, in magnetic resonance imaging studies, deep neural networks could precisely delineate brain regions[21,22]. Deep learning models have also been used to predict fluorescence images of diverse cell and tissue structures[23], as well as segment neurons on images recorded through two-photon microscopy[24], but not for wide-field images.

In this study, we developed two approaches for brain atlas alignment that either scale the atlas to a brain or re-scale the brain to fit a common atlas. For atlas-to-brain, we trained a deep learning model[25] to automatically identify cortical landmarks based on single raw fluorescence wide-field images. The predicted landmarks are then used to re-scale a reference atlas (adapted from Allen Mouse Brain Atlas)[26,27] to the brain. Another fully convolutional network U-Net[28,29] was then used to delineate the brain boundaries automatically. For the brain-to-atlas approach, our system automatically registers cortical images to a common atlas using predicted cortical landmarks. This alignment approach, while robust in the presence of anatomical landmarks, does not leverage regional patterns within functional calcium imaging data that are related to underlying structural connectivity[2,30,31]. We suggest that functional maps that represent specific spatio-temporal consensus patterns of regional activation observed using activity sensors such as GCAMP6[2,30,31] or potentially hemodynamic activation[32,33] can also be used for atlas registration. As such, we extended this anatomical alignment approach with three pipelines that can use functional sensory maps or spontaneous cortical activity. Spontaneous cortical activity was assessed by recovering regional activity motifs[34] and using them to generate motif-based functional maps (MBFMs). MBFMs can then be used to train a learning-based framework, VoxelMorph[35], which nonlinearly deforms the reference atlas to register it to the brain image. An MBFM based U-Net model (MBFM-U-Net) can directly predict positions of anatomical brain regions from the spatial structure of MBFMs. We demonstrate that this new open-source toolbox for automated brain image analysis is robust to morphological variation and can process multiple data sets in a relatively automated manner.

## Results

**Brain-to-atlas and atlas-to-brain scaling in MesoNet**. We present an open-source toolbox (https://osf.io/svztu) that will facilitate the analysis of mesoscale imaging data from wide-field microscopy. Brain-to-atlas and atlas-to-brain registration and scaling in MesoNet can help to account for differences between animals and imaging conditions which, in turn, can facilitate group data analysis or averaging functional maps between animals. For this work, we have chosen a flattened areal view of the cortex (see Fig. 1). The brain-to-atlas MesoNet approach scales each brain to a common reference atlas with predicted landmarks, such as bregma, over raw mesoscale GCaMP6 images with similar performance as human raters (see Fig. 2). Alternatively, the atlas-to-brain approach re-scales a reference atlas to fit brain data for regional analysis of brain activity. While we see brain-to-atlas scaling as being the most appropriate method for aggregating experiments, MesoNet can handle special cases such as brains that have been imaged at different angles or brains that are partly out of the frame and will return a set of best-fit regions of interest that can be matched with known anatomical regions by users.

**Landmark definition in the common coordinate system**. We first define the landmarks in a common coordinate system for alignment to the reference atlas. To determine suitable landmarks for image registration and model training, we first employed an inpainting method to process the raw images and remove cortical-tissue patterns such as blood vessels unrelated to regional borders (Supplementary Fig. 1a). We then averaged brain images (images were manually aligned during experiments, $n = 12$ mice) to determine consensus anatomical structures that fit a reference atlas (Supplementary Fig. 1b, c). We selected nine clearly defined landmarks[36,37] and created a common coordinate system while setting the skull landmark Bregma as (0,0 mm) (Fig. 1b, c; Supplementary Table 1). We see inpainting as a step in validating cortical landmarks for pre-training the models, but it is not required for a typical MesoNet pipeline.

**Robust landmark estimation using deep learning**. In order to automatically estimate landmark locations, a dataset of 491 images annotated with nine anatomical landmarks was used to train a landmark estimation network via DeepLabCut[25] ("OSF Storage/6_Landmark_estimation_model" at https://osf.io/svztu). To evaluate the performance of this deep learning model for landmark estimation, we calculated the pairwise Euclidean distance (root mean squared error: RMSE) between human annotation and model-generated labels of a testing dataset of 20 brain images. This permitted us to assess labelling precision at each landmark during training (Supplementary Fig. 2a, b). We then tested our trained network on a novel set of 20 wide-field cortical calcium images ($n = 20$ mice) and compared the results of model labelled landmarks with two different human raters. The average distance between the placement of our network and human annotators was generally around 0.1 mm and not more than the difference between human raters (Fig. 2; Supplementary Table 2).

**Raw fluorescence delimitations using U-Net**. In parallel to the landmark identification, we also developed an approach to delimit the boundaries of the cortex (below the transparent skull). This delimiting method was based on training an adapted U-Net network[29] (Supplementary Fig. 2c, d) using manually delimited regions. After training, we compared the segmentation quality of the U-Net network and a conventional segmentation method – Otsu's threshold method – for each image by using a paired-samples t-test based on four criteria: area difference, structural similarity index, peak signal-to-noise ratio, and mean squared error (Fig. 3). All of these criteria, as well as the larger differences (the green region in Fig. 3a) in segmentation, clearly showed that the U-Net model offers higher accuracy for cortical delimitation prediction compared to the conventional Otsu's threshold method (when both were compared to the human rater's manual delimitation).

**Validation of performance of atlas-to-brain alignment by sensory mapping**. For atlas-to-brain alignment, our system

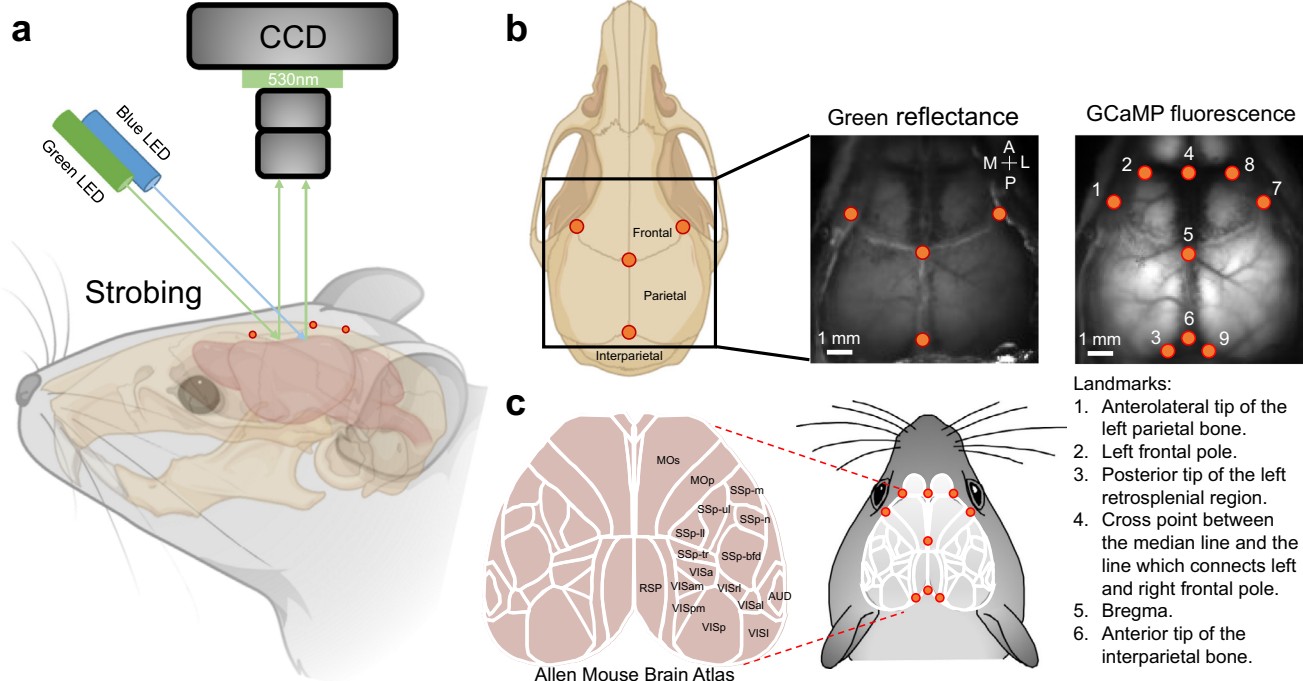

**Fig. 1 Set up of wide-field calcium imaging and definition of landmarks. a** Schematic showing green (560 nm) and blue (480 nm) LED lights targeted directly above the cranial recording window that were used to illuminate the cortex. Green reflectance and emission fluorescence were filtered using a 510–550 nm bandpass filter. The mouse head and skull were created with BioRender.com. **b** Examples of raw GCaMP and green reflectance images are shown with annotated landmarks. **c** Reference atlas (white outlines; ©2004 Allen Institute for Brain Science. Allen Mouse Brain Atlas. Available from: http://mouse.brain-map.org/) used for our segmentation process. Mop, primary motor area; Mos, secondary motor area; SSp-m, primary somatosensory area, mouth; SSp-ul, primary somatosensory area, upper limb; SSp-ll, primary somatosensory area, lower limb; SSp-n, primary somatosensory area, nose; SSp-bfd, primary somatosensory area, barrel field; SSp-tr, primary somatosensory area, trunk; VISp, primary visual area; VISa, anterior visual area; VISam, anteromedial visual area; VISpm, posteromedial visual area; VISrl, rostrolateral visual area; VISal, anterolateral visual area; VISl, lateral visual area; RSP, retrosplenial area; AUD, auditory areas.

combined the aligned atlas and segmented cortical boundaries to delimit each cortical area in each recording (Fig. 4a). We used sensory mapping to validate expected delimited cortical regions. To perform sensory mapping, we applied visual and tactile (whiskers and tail) stimulations on mice (Fig. 4b) to generate activation maps for the respective sensory modalities (Fig. 4c). We expected that the sensory stimulation paradigms would activate analogous areas of the cortex across different mice. Cortical mapping is presented in terms of relative activation (ΔF/F) and is not strictly dependent on the basal level of GCaMP calcium-induced fluorescence. Calcium responses were averaged between epochs, and the profiles of calcium fluctuations were calculated before and after sensory stimulation by determining the fluorescence (ΔF/F) time series (Fig. 4d). Then, the sensory maps were generated by calculating the maximum ΔF/F for each pixel after stimulation, and the peak activation (Fig. 4c) was considered as the functional coordinate center (Supplementary Fig. 3). The predicted sensory regions from MesoNet (nine landmarks plus U-Net) were consistent with sensory functional maps (Fig. 4c, d)[5,30].

**Testing the brain-to-atlas scaling across different lines of fluorescent protein mice.** Since the training of our networks could be influenced by specific patterns of expression of the fluorescent indicators, such as those observed in TIGRE mice[38], we validated the robustness of the method by testing other lines. We examined six different lines of mice expressing fluorescent indicators, as well as mice injected with blood-brain barrier permeable PHP.B virus allowing expression of GCaMP6[39] (GCaMP6f, $n = 4$ mice, GCaMP6s, $n = 4$ mice, GCaMP3, $n = 4$

mice, PHP.B, $n = 4$ mice, GFP[40,41], $n = 4$ mice, Thy1-GCaMP[42], $n = 4$ mice, iGluSnFr[43], $n = 4$ mice, jrGECO[44], $n = 4$ mice, Green reflectance on wild-type mice, $n = 4$ mice) (Fig. 5a, b). Without any manual delimitation of landmarks, we could register atlas overlays to all of these examples. This result indicates that the performance of MesoNet is not specific to a particular mouse line, expression profile, or wavelength. Although this performance could differ across mouse lines (due to regional promoter activity[45]), the pre-trained model can be easily fine-tuned using a small dataset of line-specific brain images and a re-training step if needed (online training, see Methods). To further quantify the performance of brain-to-atlas alignment, we compared MesoNet with manually labelled alignment by calculating the Euclidean distance between the landmarks of the anterior tip of the inter-parietal bone and cross point between the median line and the line which connects the left and right frontal pole, and angle of the midline compared to the ground truth common atlas. MesoNet performs significantly better than manual labelled alignment in both comparisons (Fig. 5c, d).

**The performance of brain-to-atlas registration during clustering cortical activity motifs.** Scaling the brain image to fit an atlas (registration) allows researchers to normalize data more efficiently and reduce the effect of brain position or angle perturbations when analyzing data. The transformation of brain images into common atlas also allows us to combine cortical maps from different animals. To evaluate the capabilities of our registration pipeline, we used seqNMF[34,46] to discover cortical activity motifs from resting-state mesoscale cortical imaging from 6 head-fixed GCaMP6s mice. This method was able to reveal spatio-temporal

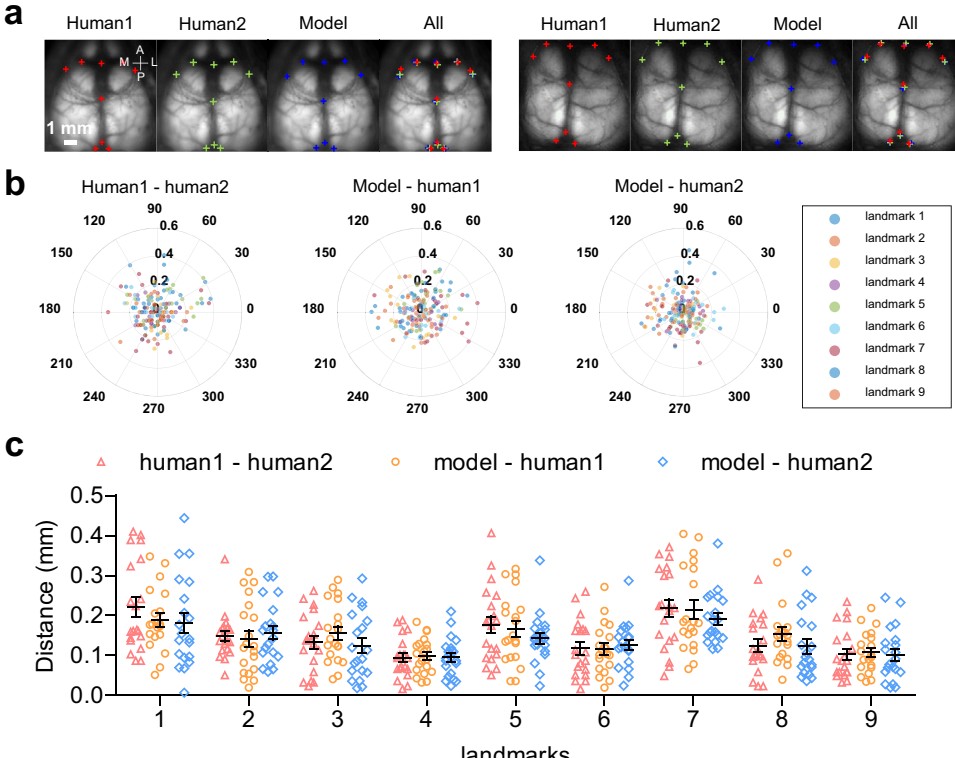

**Fig. 2 Performance of automated landmark estimation. a** Examples of model labelled and manual labelled landmarks on GCaMP images (denoted by '+' symbols; blue, model labelled; red and green, manual labelled). **b** Polar plot of distance between coordinates of model labelled and manual labelled landmarks. **c** Comparison of distance between coordinates of model labelled ($n = 20$ mice) and manual labelled landmarks ($n = 20$ mice) and distance calculated by differences between coordinates of two runs of manual labelled landmarks ($n = 20$ mice) (scatter dot plot, line at mean with SEM, Bonferroni tests (two-sided): human1 - human2 vs model - human1, $p > 0.05$; human1 - human2 vs model - human2, $p > 0.05$; model - human1 vs. model - human2, $p > 0.05$; mean distance and SEM for each landmark see Supplementary Table 2). Source data are provided as a Supplementary Data file.

cortical activity motifs (Fig. 6) that represent known intracortical connection patterns[14,30,31].

As a further test of our software, we artificially generated a more diverse, mis-aligned test dataset by rotating the brain data to arbitrary angles and resizing the images by variable factors (Fig. 6a). We then compared the motif clusters for raw data from mis-aligned data before and after brain-to-atlas transformation using an unsupervised clustering algorithm (PhenoGraph)[47]. We found that the brain-to-atlas approach was able to normalize the motif cluster number in the mis-aligned datasets (8 clusters for transformed data and 12 clusters for mis-aligned synthetic data, Fig. 6b), which apparently misclassified some motif patterns as new clusters (Fig. 6b, c). We further quantified the clusters by calculating the silhouette score (values approaching 1 indicate clusters are more separated from each other and clearly distinguished), showing a better separation after brain-to-atlas transformation. The silhouette score calculated from raw data was 0.43, from mis-aligned synthetic data was 0.39, and the score from brain-to-atlas transformed data was the highest at 0.48, indicating clusters with the least overlap.

**Alignment using spatial-temporal functional cortical activity signals**. An advantage of MesoNet is that most alignment can be performed using only a single raw fluorescent image (9 landmarks plus U-Net). In this case, MesoNet alignment is mostly dependent on cortical bone and brain edge markers and does not consider internal functional boundaries. While this approach does show good correspondence with the location of expected sensory signals (Fig. 4), it would be advantageous to also make use of functional maps to track the dynamic organization of functional

cortical modules in different sensory and cognitive processes, as well as the precise topography of brain parcellation. Previously, we and others[30,31,48] have used regional correlations of GCaMP signals during spontaneous activity to establish brain functional networks that correspond to underlying anatomical projections. While correlations yield robust maps, they do require the placement of seed locations and some underlying assumptions of anatomical mapping[31].

As a potentially less-biased approach, we employ seqNMF[34,46] (as in Fig. 6) to recover stereotyped cortical spatio-temporal activity motifs as a means of establishing functional maps. This approach generates motif patterns that only require spontaneous activity and would be advantageous over sensory modality mapping that requires specialized forms of stimulation and additional imaging trials (Figs. 4, 7a). To perform seqNMF motif recovery, an averaged mask (15 mice) was applied to limit the motif analysis to areas inside the brain window. In these experiments, brains were roughly pre-aligned during GCAMP data acquisition. As shown in Fig. 6, this approach can recover at least six major spatial-temporal activity motifs from each brain. To create an aggregate picture of motif boundaries, we scaled each motif to its maximal value and then created a summed maximal intensity projection (Fig. 7b, c).

Like previous projections of seed, pixel maps gradients[31], maximal projection of the collection of motifs that represent cortical activity led to the definition of functional "firewall" boundaries that reflect weighted activity transitions between major cortical groups of areas. Importantly, these firewalls were relatively stable across different animals where functional resting-state GCaMP activity was observed and can be used to create

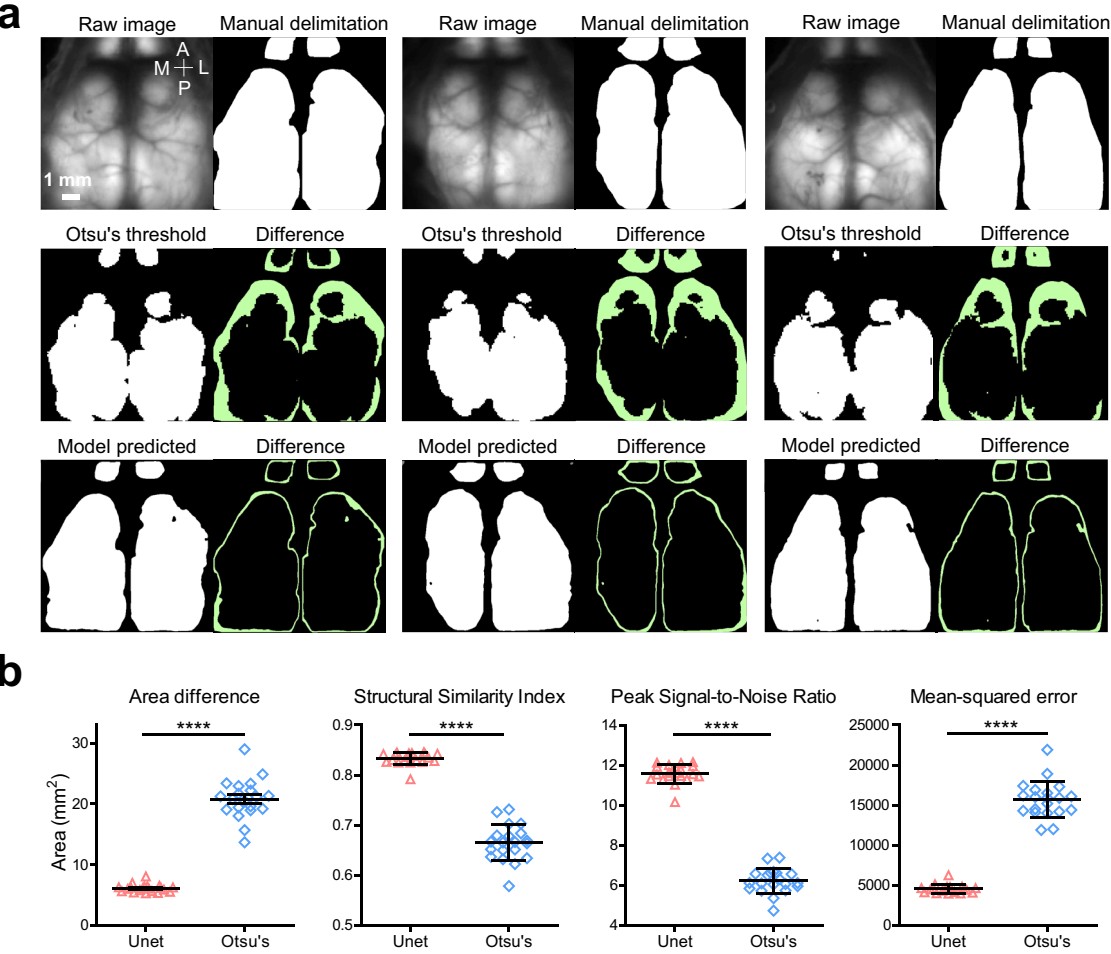

**Fig. 3 Performance of automated delineation of brain boundary using U-Net. a** Representative images showing raw GCaMP images and respective human-applied brain delimitation as ground truth. Brain delimitation predictions from application of Otsu's threshold (middle panel) and U-Net model segmentations (bottom panel). Green areas are the absolute differences between predicted versus ground truth. **b** Comparison of model-predicted ($n = 20$ mice) and Otsu's threshold ($n = 20$ mice) brain delimitation results to ground truth by mean values for area difference (scatter dot plot, line at mean with SEM, paired $t$-test (two-tailed), ****$p$ values <0.0001; U-Net, mean ± SEM = 6.11 ± 0.14; Otsu, mean ± SEM = 20.81 ± 0.73; $p < 0.0001$, $t = 23.24$), structural similarity index (U-Net, mean ± SEM = 0.83 ± 0.003; Otsu, mean ± SEM = 0.66 ± 0.01; $p < 0.0001$, t = 26.08), $p$eak signal-to-noise ratio (U-Net, mean ± SEM = 11.57 ± 0.1; Otsu, mean ± SEM = 6.22 ± 0.14; $p < 0.0001$, $t = 56.12$) and mean squared error (U-Net, mean ± SEM = 4551 ± 115.9; Otsu, mean ± SEM = 15680 ± 507.7; $p < 0.0001$, $t = 26.12$). Source data are provided as a Supplementary Data file.

animal-specific motif-based functional maps (MBFMs) (Fig. 7b, c, Supplementary Fig. 4). These MBFMs provide an opportunity to predict brain regional boundaries (represented by a cortical overlay as the output of 9 landmarks plus U-Net in Fig. 4a) using another pre-trained MBFM based U-Net model (Fig. 7b) that we call the MBFM-U-Net model.

To supplement our anatomical landmark-based alignment approach, we capture local deformation using functional map features by integrating VoxelMorph[35] as an optional add-in to the MesoNet pipeline (Fig. 7c, Supplementary Fig. 2e). VoxelMorph offers a learning-based approach that determines a deformation field that is required for the transformation and registration of image pairs such as MBFMs. We generate a template MBFM that is aligned with an anatomical reference atlas (common atlas framework, Fig. 7d). The deformation field predicted from VoxelMorph can be applied to the reference atlas to fit the functional regions in the input MBFM (atlas-to-brain).

To check the performance of these mouse-specific MBFM-based alignments, we compared the predicted location of sensory regions for sensory map-based (Fig. 7a), MBFM-U-Net (Fig. 7b), and VoxelMorph (Fig. 7c) pipelines. The prediction accuracy was

then evaluated by measuring the Euclidean distance between the centroids of sensory stimulation-induced activation and predicted atlas ROI centroids (Fig. 7e, f). All the three pipelines yielded similar distances to anatomical sensory map centres, although the VoxelMorph pipeline performed worse (in the barrel cortex BCS1 center, Fig. 7f). The VoxelMorph pipeline's performance was improved by first applying a brain-to-atlas transformation to the MBFMs (VoxelMorph transformed, Fig. 7f, BCS1). We further evaluated the performance of these pipelines by calculating the correlation coefficient between manually delineated retro-splenial (RSP) regions (RSP consistently has clear boundaries in GCaMP functional data, Fig. 7d, e, Supplementary Fig. 1c) and model-predicted RSP regions. In this case, VoxelMorph performed significantly better than other pipelines as it warped brain areas to fit functional boundaries in MBFMs (Fig. 7g, Supplementary Movie 1).

We suggest that, under certain conditions, there may be unique advantages to employing additional, computationally more-intensive steps such as combining brain-to-atlas with VoxelMorph. These conditions might include brains that were significantly rotated or shifted between the experiments (Supplementary Fig. 5),

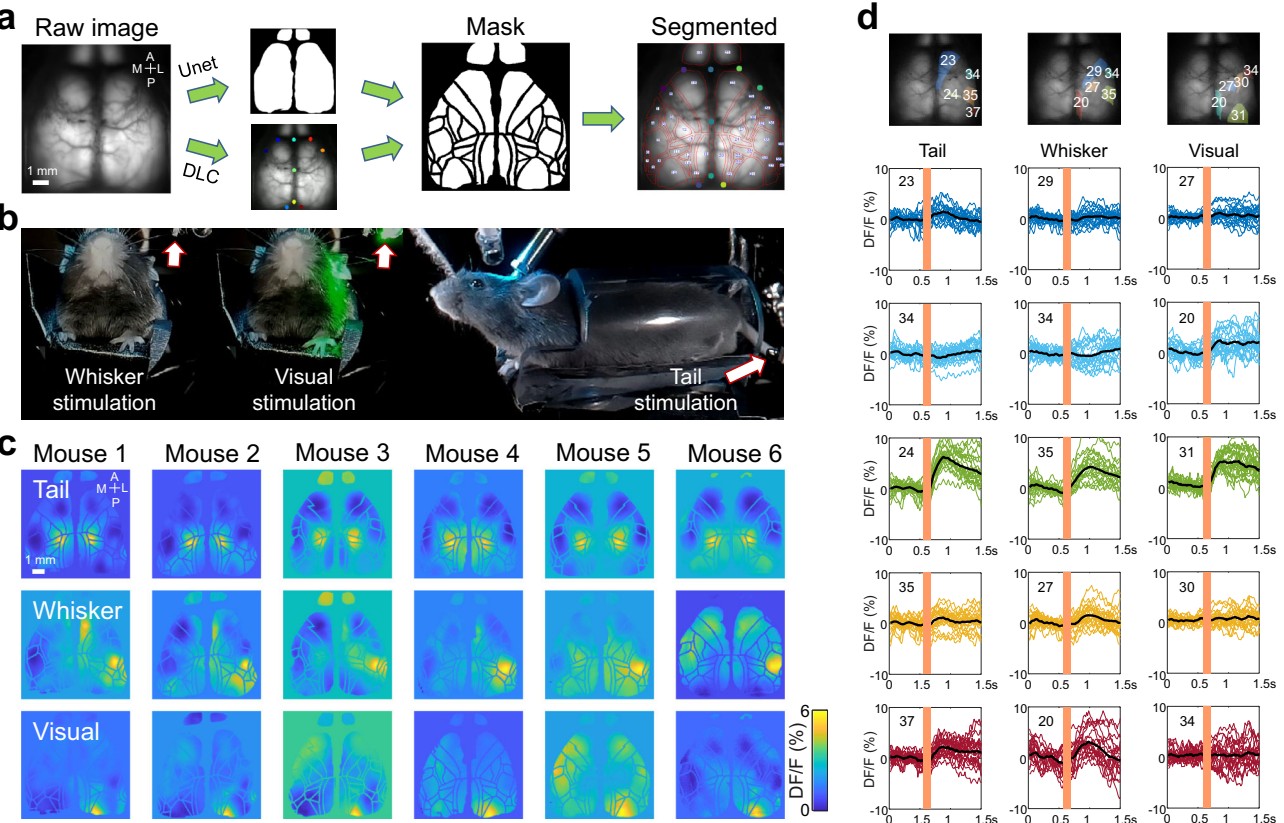

**Fig. 4 Sensory mapping in awake mice. a** Automated alignment and segmentation pipeline of the atlas-to-brain approach. The raw GCaMP image was segmented using U-Net, and landmark estimation was done using DLC and then combined together to determine each brain region ROI. **b** Frontal and lateral view of an experimental set-up involving head-fixed mice with sensory stimulation cranial recording. **c** Sensory mapping across independent trials ($n = 6$ mice) shows similar regions of activation resulting from physical stimulation of the tail or whiskers and visual field of the mouse. **d** Single trials ($n = 30$ trials) of calcium temporal dynamics around the tail, whisker, and visual stimulations of different brain regions (indicated with the same colour on the brain image, the number of brain region ROIs are automatically output from MesoNet). The black line is the averaged calcium response of all the trials.

or analyses of specific lines of mice in which phenotypes affect neuroanatomical borders, or conditions such as lesions that may make alignment to the consensus atlas more challenging. In summary, we present multiple means of registering cortical mesoscale functional images to cortical overlay atlases that are transformations of a common Allen Brain atlas, or can be extended to include animal-specific functional connectivity.

## Discussion

We present an automated, Python-based image processing pipeline for accurate alignment and segmentation of cortical areas from mesoscale cortical fluorescent images. We developed atlas-to-brain and brain-to-atlas alignment approaches using anatomical landmarks and brain boundaries. We also extend our pipeline to make use of functional sensory maps and spontaneous cortical activity motifs. Finally, we included a GUI and a command-line interface (CLI) to improve ease of use for non-specialists and those who are not as familiar with scripting-based techniques, enabling the use of our toolbox in a variety of projects and conditions.

In order to carry out image registration, we leveraged multiple sources of anatomical and functional landmarks derived from brain images as well as skull junctions. Anatomical landmarks within the skull, such as the bregma and lambda, have historically been used for decades to define locations of brain regions in rodents[7,49]. Surprisingly, we discovered that some unconventional anatomical landmarks based on the frontal poles (landmarks 2 and 8) or the posterior tip of the

retrosplenial region (landmarks 3 and 9) facilitated similar or more accurate atlas alignment than more familiar markers such as bregma (Fig. 2c). Automatic landmark estimation provided a similar average placement error than between two different human experimenters using manual annotation. High precision of image registration is required as studies in the visual system report functional activity transitions over 0.1 mm scales[2,50].

Other regional landmarks, such as intrinsic fluorescence or light reflectance, could provide valuable spatial information for the registration and would involve training for each mouse strain and imaging modality specifically. Expression of genetically encoded calcium indicators (GECIs) such as GCaMP provides clear fluorescence that allows for characterization of neuronal activity over large cortical regions. We observed that the basal GCaMP6 fluorescence in some cortical structures, such as the retrosplenial area (Supplementary Fig. 1), primary sensorimotor area, and secondary motor area, are relatively enhanced and visible after averaging. This was especially a characteristic of some GCaMP6 mouse lines developed by the Allen Brain Institute, such as TIGRE[6,38]. Since intracellular calcium levels are related to neuronal spiking and excitation are measured by wide-field GCaMP-fluorescence activity[5,51], it is possible that the enhanced fluorescence we observe when averaging could represent areas of increased neural firing or line-specific differences in GCaMP6 expression. However, this regional pattern of GCaMP6 fluorescence was not observed in other mouse line strategies such as in tetO mice[52], Thy1-GCaMP6[42], or ROSA26[5] suggesting differences in GCaMP expression. Importantly, MesoNet works equally

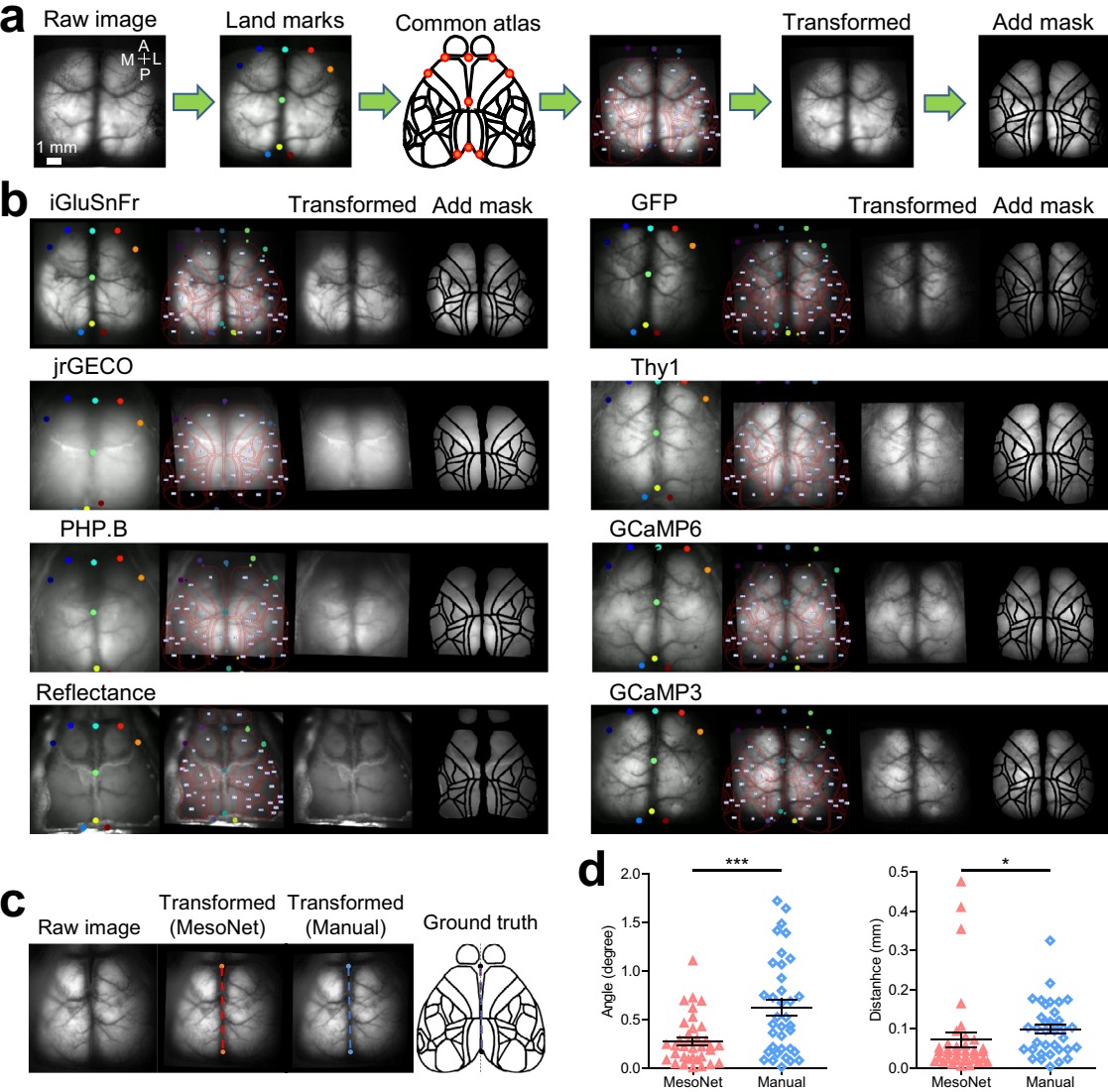

**Fig. 5 Testing the brain-to-atlas approach across different lines of fluorescent protein mice. a** Automated registering and scaling pipeline of brain-to-atlas transformation (the fourth panel: big dots denote predicted landmarks on raw image, small dots denote landmarks on a common atlas, red line denote segmented brain regions; ©2004 Allen Institute for Brain Science. Allen Mouse Brain Atlas. Available from: http://mouse.brain-map.org/). **b** Example results of the brain-to-atlas transformation of brain image from different transgenic or virally injected mice. **c** Example images showing brain to atlas alignment using MesoNet or manually labelled landmarks (bregma, window margins). The performance of the alignment is based on the calculation of the distance between landmarks of anterior tip of the interparietal bone and cross point between the median line and the line which connects the left and right frontal pole, and angle of the midline compared to the ground truth common atlas, and angle of the midline compared to the ground truth common atlas (all distances and angles are reported as positive deviations compared to ground truth common atlas). **d** Distribution of the angle (scatter dot plot, line at mean with SEM, ***$p$ values < 0.001, * denote $p$ values <0.05; MesoNet, mean ± SEM = 0.28 ± 0.04; Manual, mean ± SEM = 0.62 ± 0.08; Wilcoxon signed-rank test, Two-tailed, $p = 0.0005$, Sum of signed ranks (W) = −446, $n = 36$ mice) and distance (MesoNet, mean ± SEM = 0.07 ± 0.02; Manual, mean ± SEM = 0.1 ± 0.01; Wilcoxon signed-rank test, Two-tailed, $p = 0.0122$, Sum of signed ranks (W) = −320, $n = 36$ mice). MesoNet performs significantly better in both comparisons. Source data are provided as a Supplementary Data file.

well with mouse lines with these different regional profiles of GCaMP6 expression (see below).

The current approach was also tested and worked well on other lines of GECI mice, iGluSnFr, GFP, Green reflectance on wild-type mice, as well as mice injected with a blood-brain barrier permeable PHP.B virus expressing GCaMP6. Given this versatility, it is reasonable to believe that the methods presented here could work on any strain of fluorescent protein-expressing mouse or rat, as well as other indicators (e.g., iGluSnFr in Ai85 mice) or optogenetic fusion proteins (e.g., Thy1-ChR2-YFP) or in non-transgenic animals with brain-wide infection by viruses such as PHP.B[41,53].

Other pipelines, such as brainreg[54], have been developed to facilitate automatic registration of various types of brain imaging data based on common structures (such as gyri, sulci, shape, and edges) between images[35,54]; however, these pipelines are primarily oriented towards aligning images with distinct anatomical features that remain relatively consistent across brains. We tested brainreg on our fluorescent images but did not produce consistent transformations. We determined that the feature-matching procedure utilized by brainreg requires a close correspondence in features between brains. Such anatomical features are present and consistent across many types of histological brain slice data, but not in raw

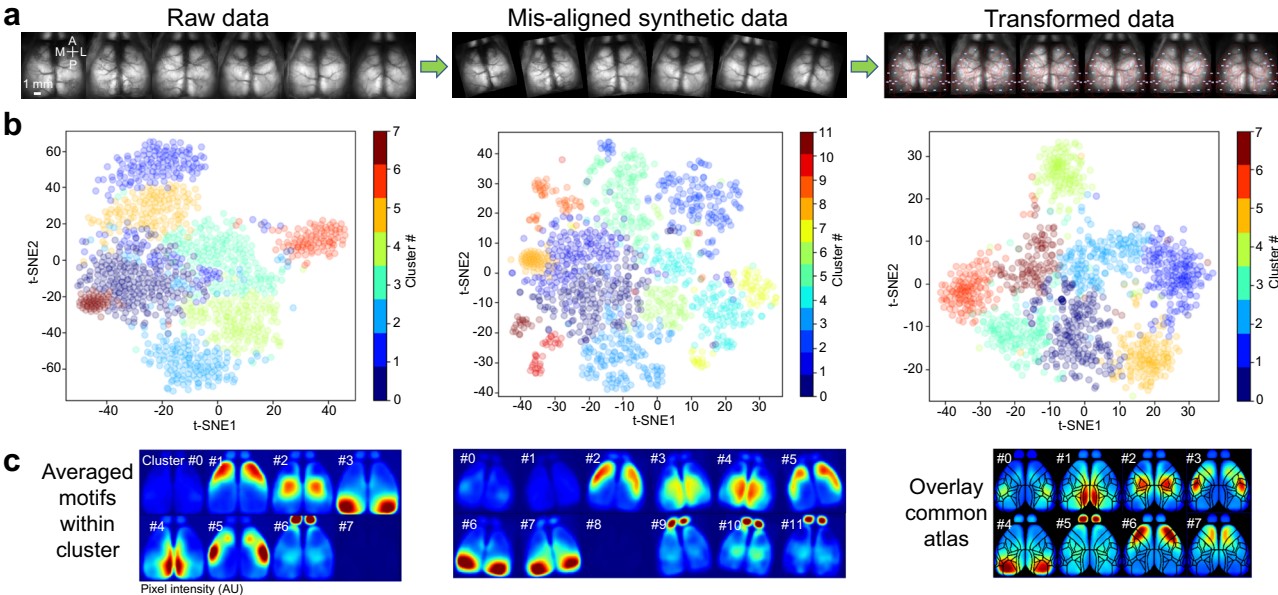

**Fig. 6 Performance of brain-to-atlas transformation for clustering cortical activity motifs. a** Raw brain images, synthetic brain images, and brain-to-atlas transformed brain images from 6 GCaMP6 mice. **b** Scatter plot of motif clusters. An unsupervised clustering algorithm (Phenograph) was used to classify the motifs ($n = 1194$ motifs from 6 mice). Different colors in the t-SNE plot indicate different motif clusters. Left panel: scatter plot of motif clusters of 6 mice using raw data, silhouette score = 0.43. Middle panel: motif clusters of synthetic misaligned data, Silhouette Score = 0.39. Right panel: motif clusters of transformed data, silhouette score = 0.48. **c** Averaged motifs for each cluster of raw, synthetic mis-aligned, and transformed data. All the spatio-temporal motifs in each cluster were averaged, and the maximum temporal dynamics were projected onto one image. The pixel intensity scale is normalized, and the intensity value is arbitrary because responses are convolved with independently scaled temporal weightings to reconstruct the normalized DF/F fluorescence.

fluorescence or functional data such as that derived from mesoscale calcium images. Furthermore, vasculature - a highly visible feature across many calcium imaging datasets - is mostly unique across mice, providing further visual differentiation between images that challenge such feature-matching approaches to image registration. As such, brainreg is primarily useful for registering anatomical images - such as histological brain slices - whereas MesoNet offers a novel machine learning-based approach to robust functional image registration.

To increase the flexibility and accessibility of MesoNet, we developed a landmark-based pipeline (Fig. 7a), an MBFM-U-Net-based pipeline (Fig. 7b), and a VoxelMorph-based pipeline (Fig. 7c). We also provided six end-to-end automated pipelines[55,56] to allow users to quickly output results from input images (Supplementary Fig. 6 and Supplementary Movie 2). The landmark-based approach is most advantageous when landmarks are visible in the brain images (our pipeline is flexible to employ a different number of landmarks to align the brain) and when brain images are rotated or shifted. However, its registration procedure relies on a limited number of landmarks, and an overlay atlas may not capture local deformation that could be accounted for using methods that employ deformation fields such as VoxelMorph. The sensory map-based approach has the advantage of being able to utilize ground truth sensory induced activation areas, but this approach relies on experimental expertise and is susceptible to errors in the placement of the stimulation devices. The MBFM-U-Net approach is most useful for data with distinct functional features but non-distinct anatomical landmarks. However, the model training of MBFM-U-Net is supervised and needs a well-aligned label for each brain image, and its effectiveness may be reduced if contrast and image features differ significantly from the training dataset. The VoxelMorph approach provides a fast learning-based (unsupervised) framework for deformable registration, but it is less robust to larger rotations or shifts in cortical position in the frame (Supplementary

Fig. 5). In order to address each dataset's individual strengths and weaknesses, MesoNet allows anatomical and functional approaches to be combined. We suggest first applying brain-to-atlas alignment using a landmark-based pipeline then combining this with VoxelMorph as a better option to align and deform a reference atlas to functional maps (Supplementary Fig. 6f).

Overall, we apply machine learning models to automate the registration and overlay of the reference atlas and the segmentation of brain regions using mesoscale wide-field images with high accuracy. We developed animal-specific motif-based functional maps that represent cortical consensus patterns of regional activation that can be used for brain registration and segmentation. Our automated pipelines can be combined to consider both anatomical consistency and functional individual variations to help better analyze brain regional activity. Our open-source platform, MesoNet, allows researchers to register their functional maps to a common atlas framework based on cortical landmarks and will help comparisons across studies.

## Methods

**Animals and surgery.** Animal protocols (A18-0036 and A18-0321) were approved by the University of British Columbia Animal Care Committee and conformed to the Canadian Council on Animal Care and Use guidelines. Animals were housed in a vivarium on a 12 h daylight cycle (7 AM lights on), with controlled room temperature at $24 \pm 2 °C$ and relative humidity at 40-50%. Most experiments were performed towards the end of the mouse light cycle. Transgenic GCaMP6f, GCaMP6s or iGluSnFR mice (males, 2–4 months of age, weighing 20–30 g), were produced by crossing Emx1-cre (B6.129 S2-$Emx1^{tm1(cre)Krj}$/J, Jax #005628), CaMK2-tTA (B6.Cg-Tg(Camk2a-tTA)1Mmay/DboJ, #007004) and TITL-GCaMP6f (Ai93; B6;129S6-$Igs7^{tm93.1(tetO-GCaMP6f)Hze}$/J, #024103) or TITL-GCaMP6s (Ai94;B6.$Cg$-$Igs7^{tm94.1(tetO-GCaMP6s)Hze}$/J, Jax #024104) or TITL-iGluSnFR (B6;129S-$Igs7^{tm85.1(tetO-gltI/GFP*)Hze}$/J, #026260) strain[6]. GCaMP3 mice were crossing between Emx1-cre and B6.Cg-$Gt(ROSA)26Sor^{tm38(CAG-GCaMP3)Hze}$/J (#029043) strains. GFP, Thy1, and jrGECO mice were single strain of Tg(Thy1-EGFP)MJrs/J (#007788), C57BL/6J-Tg(Thy1-GCaMP6s)GP4.3Dkim/J (#024275), and STOCK Tg(Thy1-jRGECO1a)GP8.20Dkim/J (#030525). Finally, some wild-type mice (PHP.B) were IV injected with virus crossing the blood-brain barrier and allowing the expression

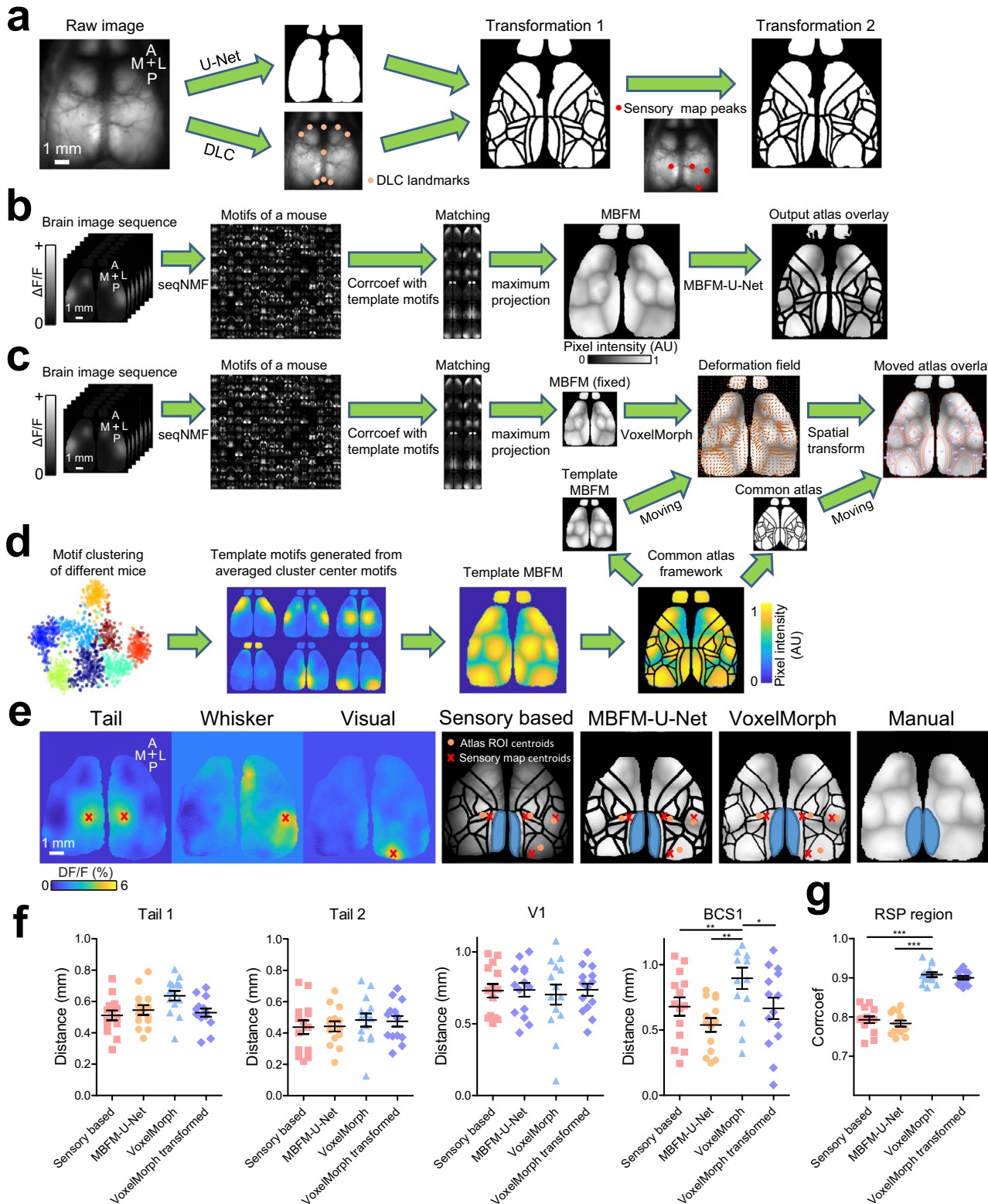

of the indicator everywhere in the cortex (AAV.PHPeB.Syn.GCaMP6s)[57]. The presence of indicator expression was determined by genotyping each animal before each surgical procedure with PCR amplification.

For the chronic window surgery, animals were anesthetized with isoflurane (2% in pure $O_2$), and the body temperature was maintained at 37 °C using a feedback-regulated heating pad monitored by a rectal thermometer. Mice received an intramuscular injection of 40 µl of dexamethasone (2 mg/ml) and a 0.5 ml subcutaneous injection of a saline solution containing buprenorphine (2 µg/ml), atropine (3 µg/ml), and glucose (20 mM) and were placed in a stereotaxic frame.

After locally anesthetizing the scalp with lidocaine (0.1 ml, 0.2%), the skin covering the skull was removed and replaced by transparent dental cement and a glass coverslip[7]. A metal screw was attached to the chamber for future head fixation during recordings.

**Wide-field calcium imaging**. A Pantera 1M60 CCD camera (Dalsa) was equipped with two front-to-front lenses (50 mm, f ¼ 1.4 : 35 mm, f ¼ 2; Nikon Nikkor) and a bandpass emission filter (525⁄36 nm, Chroma). The 12-bit images were captured at

**Fig. 7 Performance of functional sensory map and activity-motif alignment pipelines. a** Sensory map-based alignment by combining nine landmarks plus U-Net pipeline with functional sensory maps (tail, visual, and whisker stimulation-induced peak activation) to align the reference atlas to brain image. **b** A spontaneous activity motif matching procedure was used to generate motif-based functional map (MBFM) using calcium imaging data detected by seqNMF. The MBFM was then used to predict brain regional boundaries using MBFM based U-Net model (MBFM-U-Net). **c** The MBFM is used to predict a deformation field corresponding to a template MBFM using VoxelMorph. The deformation field will then be applied to the reference atlas to fit input MBFM. **d** The generation of template MBFM (see Methods). Different colors in t-SNE plot indicate different motif clusters. **e** Example images show the sensory maps and output atlas from sensory-based, MBFM-U-Net and VoxelMorph pipelines and manually painted RSP region on MBFM (blue). **f** Comparison of the performance of the pipelines (sensory-based, MBFM, VoxelMorph, and VoxelMorph after brain-to-atlas transformation of the MBFMs) by calculating the euclidean distance between the centroids of sensory stimulation-induced activation and predicted atlas ROIs (Tail, V1, and BCS1). Scatter dot plot, the line at mean with SEM, one-way ANOVA with Dunn's Multiple Comparison Test. ***$p < 0.001$, **$p < 0.01$, *$p < 0.05$; Sensory-based vs. Voxelmorph, $p < 0.05$, rank $= -22$; MBFM-U-Net vs Voxelmorph, $p < 0.05$, rank $= -25$; VoxelMorph vs VoxelMorph transformed, $p < 0.05$, rank $= 21$, $n = 14$ mice). **g** Comparison of the performance of the pipelines (sensory-based, MBFM-U-Net, VoxelMorph and VoxelMorph after brain-to-atlas transformation) by calculating the correlation coefficient between manually painted RSP region (ground truth) and predicted RSP region by different pipelines. VoxelMorph performed significantly better than other pipelines (Sensory based vs VoxelMorph, $p < 0.05$, rank $= -29$; MBFM-U-Net vs VoxelMorph, $p < 0.05$, rank $= -33$; Sensory based vs VoxelMorph transformed, $p < 0.05$, rank $= -23$; MBFM-U-Net vs VoxelMorph transformed, $p < 0.05$, rank $= -27$, $n = 14$ mice). Source data are provided as a Supplementary Data file.

---

a frame rate of 120 Hz (exposure time of 7 ms) with $8 \times 8$ on-chip spatial binning using EPIX XCAP V3.8 imaging software. These imaging parameters have been used previously for voltage-sensitive dye imaging[30] as well as anesthetized GCaMP3 imaging of spontaneous activity in the mouse cortex[5] and awake GCaMP6-imaging in the mouse cortex with chronic window[7]. The cortex was sequentially illuminated with alternating blue and green LEDs (Thorlabs). Blue light (473 nm) with a bandpass filter (467 to 499 nm) was used to excite calcium indicators, and green light (525 nm) with a bandpass filter (525/50 nm) was used to observe changes in cerebral blood volume in alternating images. The blue and green LEDs were sequentially activated and synchronized to the start of each frame's exposure period with transistor–transistor logic such that each frame collected only fluorescence or reflectance signals at 60 Hz each. Images of reflectance, used for blood artifact corrections, were also evaluated in the current pipeline (see Fig. 5 and Supplementary Table 3).

**Sensory stimulation in awake mice**. In order to validate automatic delineation of cortical maps, we applied a sensory stimulation paradigm on the tail, whiskers, and eyes in awake mice (Fig. 4b). To stimulate the tail, a mini vibration motor (weight: 2 g, dimensions: $12 \times 6 \times 3.6$ mm) was attached to the tail, and a 0.2 s vibration pulse was delivered at 10–55 Hz with an inter-stimulus interval of 10 s. To stimulate the upper part of the whiskers field, another mini-vibrator was attached to a mini-brush and given a single 0.2 s tap using a square pulse. To deliver visual stimulation, a 2 ms flash of combined green and blue light was displayed in a consistent way between animals (position 20 mm from the left eye, azimuth: 90 deg from the axis of the animal, elevation: 0 deg). Responses from 20–40 trials were averaged for each sensory stimulation, and the maximum variation of fluorescence ($\Delta F/F$) was calculated for each pixel.

**Image inpainting**. To determine whether the wide-field basal fluorescence pattern contains adequate structural information for landmark annotation (Supplementary Fig. 1), we employed an inpainting method to process the raw images and remove cortical-tissue independent patterns such as blood vessels. To generate masks of blood vessels on wide-field calcium images, we used an adaptive thresholding method implemented in OpenCV. Next, we used a Fast Marching Method algorithm[58] to fill in the region, starting from the boundary of this region. The algorithm operates by taking a region of pixels around the pixel on the area to be inpainted, then replacing the original pixel by a normalized weighted sum of all the known pixels in the neighborhood of the pixels.

**Overview of image analysis pipeline**. We developed two approaches to automatically define cortical regions: atlas-to-brain and brain-to-atlas approaches. The reference atlas was adapted from 2D cortical surface maps of the Allen Mouse Brain common coordinate framework (http://download.alleninstitute.org/publications/allen_mouse_brain_common_coordinate_framework/)[26,59,60]. In the first approach, we developed methods to re-scale the reference atlas to cortical image based on estimated cortical landmarks using deep neural networks (DeepLabCut)[25]. An adapted version of the U-Net network[29] was used to delimitate the brain boundaries automatically. Our pipeline then combines the re-scaled brain atlas and delineated brain boundary to determine brain regions. In the second approach, our system automatically re-scales and registers cortical images to our common atlas (Fig. 1, Supplementary Fig. 3, Supplementary Table 1) combined with brain boundaries (the output from U-Net) to segment brain regions.

**Pre-trained model for landmark estimation**. To make the atlas-to-brain and brain-to-atlas pipeline automatic, we developed a pre-trained model to estimate cortical landmarks on raw brain images. To generate the training dataset ("OSF

Storage/5_Model_Training_set" at https://osf.io/svztu), we randomly selected 402 wide-field cortical images from our database of experiments (Supplementary Table 3). To minimize the number of animals used in research, we used calcium imaging data that had already been collected from multiple previous and ongoing studies[7,14,61] as well as current data. These images were blindly randomized to test or train sets. A human annotator manually placed markers on nine specific anatomical landmarks to label data for the model, as shown in Fig. 1 and Supplementary Table 1. These landmarks are clearly visible and considered to be reference points for brain-to-atlas alignment[36,37]. Then, we used the labelled images to train a deep neural network (DeepLabCut)[25,62] with a 95% training and 5% testing split. The pre-train neural network feature detector architecture relies on deep residual networks (in our case, ResNet-50) with 50 layers and initialized with weights trained on ImageNet[63,64]. Approximately 10,000 iterations were sufficient for the loss to converge when training the network on our computer (Windows 10, 64GB of RAM, 3.3 GHz, and an Nvidia Titan Xp GPU). Every 2000 iterations, snapshots of the weights were stored in TensorFlow[65].

**Automatic cortical image segmentation**. We used an adapted U-Net network[28,29] to segment cortical boundaries from fluorescent mesoscopic images based on examples defined by the user. The training dataset of the U-Net model was 60 pairs of GCaMP images associated with manually painted masks of brain boundaries. To reduce the number of training data needed, we use a data augmentation strategy implemented in Keras (https://github.com/fchollet/keras), using simple transformations as summarized in Supplementary Table 4. The U-Net network used in this study consists of ten repeated applications of $3 \times 3$ convolutions, each followed by a rectified linear unit (ReLU) activation function. A $2 \times 2$ max-pooling operation with a stride of two in the contracting path halves the resolution of the feature map and additionally doubles the number of channels. In the expanding path, repeated application of up-sampling by a factor of 2 halves the number of channels. Corresponding feature maps from the decoder are concatenated with cropped feature maps from the encoder. The final convolutional layer generates masks. The network was trained for 1000 epochs with a batch size of 30.

After training, for each image of the testing dataset, the boundaries of the cortical region were automatically defined based on the fluorescence pattern using the U-Net network. For each fluorescence image, the output was a binary mask of the same size, corresponding to the prediction of each pixel to be segmented as "non-brain" or "brain".

**Automatic atlas-to-brain and brain-to-atlas alignment**. Predicted landmarks were used to align and overlay a reference brain atlas (Allen Mouse Brain Atlas) on each brain image. For each raw GCaMP image used as an input, we first extracted the predicted locations of each cortical landmark using the pre-trained landmark estimation model. To align the brain atlas with the brain image, the cv2.getAffineTransform method was used to calculate two independent transformations. The first transformation registered the left cortical landmarks on the reference atlas (stored in a MATLAB array adapted from the Allen Mouse Brain Atlas) to corresponding landmarks labelled on the left hemisphere of the brain image. The second transformation registered the right cortical landmarks to corresponding landmarks labelled on the right hemisphere of the brain image. We then use cv2.warpAffine to warp each hemisphere of the atlas independently using a three-point transformation based on these calculations. We can use the three-point combination that the model most accurately predicted (based on the output of the sigmoid activation function by TensorFlow via DeepLabCut[25]). We can also use three default points in each hemisphere: the leftmost or rightmost point in each hemisphere, followed by the top central landmark and lambda. If these points were manually un-selected as part of the analysis, the first three points selected in the left

hemisphere and the last three points selected in the right hemisphere are used for the transformations. This arrangement offers the ability to flexibly select different sets of landmarks with which to align the atlas.

For flexibility of use (especially in cases where a brain image only offers a limited number of landmarks), our software can also complete a registration using two landmarks (using a cv2.estimateAffinePartial2D registration), three landmarks (using a single cv2.getAffineTransform registration), or four to eight or more than nine landmarks (using the method described for nine landmarks). If landmarks are unavailable, but a VoxelMorph[35] local deformation model is provided along with a template for the desired alignment of the image, then VoxelMorph can be used in place of DeepLabCut for atlas registration (see "Alternative pipelines").

Furthermore, our software allows users to register the brain image to the standard brain atlas using a similar procedure to atlas-to-brain transformation. We first apply atlas-to-brain to the brain image so each hemisphere can be masked and transformed independently. We then register each masked hemisphere to the common atlas independently using landmarks on each hemisphere. We used U-Net to segment the olfactory bulbs from the original brain image; the same alignment is then applied to these olfactory bulbs to align with the registered brain hemisphere.

**Output brain region ROIs**. The final step of our pipeline combines the registered brain atlas with the brain boundaries determined through U-Net and output brain region ROIs automatically. In order to identify and label individual brain regions on our source brain images, we applied one iteration of spatial dilation to denoise the output ROIs using OpenCV's cv2.dilate function. We segment each brain image in our dataset into regions by identifying the contours of each brain region using OpenCV's cv2.findContours function. Furthermore, we identify the centre of each contour by locating the contour's pole of inaccessibility (the most distant point from the edges of the contour) using an iterative grid-based algorithm as adapted by the python-polylabel package (https://github.com/Twista/python-polylabel). As many of the brain regions have highly irregular shapes, the centre of gravity of the contour may be outside of the contour borders, and small deviations in the contour width can lead to highly eccentric centre points. Therefore, an algorithmic identification of the pole of inaccessibility ensures that the point represents a useful metric of the centre of the contour. As the contours may not be numbered in a consistent order by default, we offer two methods to increase the consistency of labels between brains. In the first method, we align a matrix in which each brain region is filled with a unique number using the same transformations that are applied to the original binary atlas; we then draw contours that match each successive unique number (using cv2.inRange), ensuring that each brain region is uniquely associated with a label that is applied consistently across images. In the second method, we automatically number the contours from the contour with the top-leftmost centrepoint to the bottom-rightmost centrepoint, separately for each hemisphere of the functional brain image (left and right side of the bregma landmark). If the centre points of two contours are vertically aligned (i.e., within 5 px horizontally of each other), these aligned contours are consistently re-sorted from the top-most to the bottom-most centrepoint to improve the consistency of contour numbering across images.

**Alternative pipelines**. Depending on the format and contents of the data, one can select from different registration strategies to the one described above. Three strategies are available if the brain images used do not have clearly defined anatomical landmarks (necessary for the landmark-based affine transformation). First, the alignment approach can leverage cross-image commonalities in sensory-related activity. In our sensory map-based pipeline, MesoNet can detect four unilateral or bilateral sensory activation peaks and use them as control points to transform reference atlas (PiecewiseAffineTransform, a non-linear piecewise affine transformation implemented in scikit-image[66]). This additional step allows sensory stimulation activation (e.g., of the tail, whiskers, and visual field; Fig. 4c) to be used for registration based on functional and not just anatomical information in the data (Fig. 7a).

Second, one can train a U-Net model on a set of motif-based functional maps (MBFMs, see "Generation of MBFM") and label images (we used manually modified output masks from 9 landmarks plus U-Net MesoNet as label images, Fig. 4a), and then use this MBFM-U-Net model to directly predict brain regional boundaries on a new set of MBFMs. Third, one can train an unsupervised VoxelMorph model[35] on a set of MBFMs (not requiring any label data for each MBFM image to train the model). After training, the VoxelMorph model can predict a deformation field between pairs of MBFMs. The deformation field can be used to transform template MBFM to each input MBFM. Specifically, using VoxelMorph terminology, the input MBFM is fixed, and the template MBFM is the moving image (Supplementary Fig. 2e, Supplementary Fig. 5). As the template MBFM is aligned with the reference atlas (Fig. 7d), we can apply the same deformation field to transform the reference atlas to fit the input MBFM. The deformation field is also exported for optional re-use in other analyses.

**Mesoscale cortical activity motif analysis and clustering**. We used the seqNMF algorithm[46] to discover spatio-temporal sequences in wide-field calcium imaging

data. This method employs convolutional non-negative matrix factorization (CNMF) with a penalty term to facilitate the discovery of repeating sequences. To classify the cortical motifs, we used an unsupervised clustering algorithm called PhenoGraph[47]. Each motif matrix is partitioned into clusters by a graph that represents their similarity. The graph is built in two steps. First, it finds $k$ nearest neighbors for each motif (using Euclidean distance), resulting in N sets of $k$ nearest neighbors. In the second step, a weighted graph is built such that the weight between nodes depends on the number of neighbors they share. We then perform Louvain community detection[67] on this graph to partition the graph that maximizes modularity.

Specifically, mice were imaged for about 60 min each, and each recording was divided into 5-min epochs to discover spatio-temporal motifs in neural activity. After classifying the cortical motifs, we average the motifs in each cluster and project the maximum temporal dynamics onto one image (see the cortical patterns in Fig. 6c).

**Generation of MBFM**. In order to generate the template MBFM for the MBFM-U-Net or VoxelMorph approach, motifs (recovered from seqNMF) from different mice are first normalized (after brain-to-atlas transformation) to a range of 0 to 1 and clustered (Phenograph[47]). The center motifs of each cluster are then averaged to generate template motifs (Fig. 7d, $n = 6$ mice). We selected the six most common motif patterns as template motifs for the motif matching procedure (see below). These template motifs are used to generate a template MBFM using maximum projection. The template MBFM is aligned to the common atlas during the first step's brain-to-atlas transformation (Fig. 7d).

The template motifs are used to match motifs generated from new mice (Fig. 7b, c). Specifically, each new motif was first normalized to the range of 0 to 1. One can also first do a brain-to-atlas alignment using MesoNet if the new motifs are significantly rotated or shifted. We then calculated the correlation coefficient between each pair of template motifs and new motifs. A threshold was set to match most similar new motifs with template motifs. We averaged the matched motifs for each motif pattern. The new MBFM was then generated by these averaged motifs using maximum projection.

**User interfaces**. In order to make our methodology more accessible to a variety of users, we have developed a graphical user interface (GUI) that allows users to input the path to a folder containing brain images to be segmented (or a single TIFF image stack) and the segmentation model to be used. The GUI then saves and displays brain images that have been segmented and labelled based on the model's predictions, with the option to export each predicted brain region as a region of interest in a .mat file for use in MATLAB. In a potential use case, one could input a sample of brain images from one's datasets and then use the segmented output data to easily focus analyses of brain imaging data on specific brain regions. The interface also allows the user to align and overlay a brain atlas onto the image automatically. A similar GUI also allows users to train new U-Net and DeepLabCut models in future MesoNet analyses. MesoNet also offers a Python package-based command-line interface for increased flexibility and the ability to integrate MesoNet analyses into larger analytical pipelines. Demo videos (Supplementary Movies 2, 3, 4, 5, 6, and 7) are available on the OSF repository to demonstrate the user interfaces (see "OSF Storage/2_Supplementary_Movies" at https://osf.io/svztu).

**Model online training and data augmentation**. To make our system more flexible and robust to different datasets, we developed an online training and data augmentation algorithm in MesoNet. One can add additional image-mask pairs to an existing model and train the existing model with these additional images, potentially improving the robustness of an already-useful model. To facilitate this process, we leverage Keras's data augmentation features to enable automatic augmentation of all images and marks that are used to train the U-Net model. Together with DeepLabCut's image augmentation features[25] (as well as a custom function for post-hoc augmentation of existing datasets labelled in DeepLabCut), these image augmentation processes help users improve the robustness of their models to noisy or distorted data[68]. Demo videos (Supplementary Movies 8, 9, and 10) and a user's manual are available to demonstrate the model online training and data augmentation (see "OSF Storage/2_Supplementary_Movies" at https://osf.io/svztu).

**Statistics and reproducibility**. Data were analyzed using GraphPad Prism 6 and custom-written software in MATLAB R2020a and Python 3.7.9. We used a Bonferroni test to compare the distance between coordinates of model labelled and manual labelled landmarks and a paired t-test (two-tailed) to compare model-predicted and Otsu's threshold brain delimitation results on brain images from different mice. We used a Wilcoxon signed-rank test to compare the performance of the brain-to-atlas alignment using MesoNet or manual labelled landmarks, and one-way ANOVA with Dunn's Multiple Comparison Test to compare the performance of functional alignment pipelines. For the comparison of the distance between coordinates of model labelled and manual labelled landmarks, experiments were repeated 20 times. For the comparison of model-predicted and Otsu's threshold brain delimitation, experiments were repeated 20 times. For the

comparison of brain-to-atlas alignment, experiments were repeated 36 times. For the comparison of functional alignment pipelines, experiments were repeated 14 times. All attempts at replication were successful.

**Reporting summary**. Further information on research design is available in the Nature Research Reporting Summary linked to this article.

## Data availability

Raw wide-field calcium imaging data generated in this study have been deposited in the Open Science Framework (OSF) public repository (https://osf.io/34uwj). More imaging data and corresponding annotations or masks used for model training (landmark estimation model, U-Net model, MBFM-U-Net model, and VoxelMorph model), source data for figures, and demo data with code have been deposited in another public OSF repository (https://osf.io/svztu). Example brain images for testing the landmark estimation model, U-Net model, MBFM-U-Net model, and the VoxelMorph model, as well as data for the demo videos, are available on the public OSF repository ("OSF Storage/0_Example_data"; https://osf.io/svztu). Source data are provided with this paper.

## Code availability

The Python software package, pre-trained landmark estimation model, U-Net model, user's manual, and sample data for the demonstration of MesoNet are available on the public OSF repository (https://osf.io/svztu), which also offers a link to the GitHub repository[69] from which the Python package can be installed. We also provide a Google Colaboratory notebook for a fully functional version of the MesoNet command line interface within this GitHub repository (mesonet_demo_colab.ipynb). We also provide demo Matlab code and data to demonstrate the procedures to generate functional maps from spontaneous cortical activity motifs (see "OSF Storage/4_Data_code" at https://osf.io/svztu). Lastly, we provide a Code Ocean capsule to demonstrate the operation of all automated MesoNet pipelines[55] at 10.24433/CO.1919930.v1 [https://doi.org/10.24433/CO.1919930.v1], and another capsule to demonstrate the MBFM generation process[56] at 10.24433/CO.4985659.v1 [https://doi.org/10.24433/CO.4985659.v1].

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

## Acknowledgements

This work was supported by Canadian Institutes of Health Research (CIHR) Foundation Grant FDN-143209 to T.H.M., THM is supported by the Brain Canada Neurophotonics Platform, a Heart and Stroke Foundation of Canada grant in aid, and a Leducq Foundation grant. D.X. was supported in part by funding provided by Brain Canada, in partnership with Health Canada, for the Canadian Open Neuroscience Platform initiative. This work was supported by resources made available through the Dynamic Brain Circuits cluster and the NeuroImaging and NeuroComputation Centre at the UBC Djavad Mowafaghian Centre for Brain Health (RRID SCR_019086). We thank Pumin Wang, Cindy Jiang for surgical assistance; we thank Pankaj Gupta, Nicholas Michelson, Edward Yan, Rene Tandun, Jamie D Boyd, and Jeffrey M LeDue for technical assistance. We thank Hongkui Zeng and Allen Brain Institute for providing transgenic mice.

## Author contributions

D.X. contributed to conception and design of the work; collection and analysis of data; creation of new software used in the work; and drafting, writing, and revising the work. B.J.F. contributed to the design of the work, creation of new software used in the work; and drafting and revising the work. M.P.V. contributed to conception and design of the work; and drafting and revising the work. T.H.M. contributed to conception, design of the work, and writing and editing of the paper.

## Competing interests

The authors declare no competing interests.
