## [Peer Review File · Nature Communications]

MesoNet allows automated scaling and segmentation of mouse mesoscale cortical maps using machine learningReviewers' Comments:

Reviewer #1:

Remarks to the Author:

In this work Xiao and colleagues present a software package they have written to automatically delineate regions of dorsal cortex and align them to an atlas. I should preface my comments by stating that there is the possibility that I am quite fundamentally misunderstanding what the authors have done here (in which case I apologize, but would recommend to substantially rewrite the paper). I agreed to review the paper, having read the abstract, assuming the authors had developed a method to segment cortical areas based on calcium activity data (i.e. using the temporal activity patterns to delineate area boundaries). Having read the full paper, I am under the impression that they only use a single image per mouse and simply align this to an atlas (i.e. perform a slightly distorted affine transformation on a single image). The former would have been interesting and innovative. The latter is technically trivial and not worth writing a paper about. Having said that, I assume, if the software package is professionally written and supported, it will find use (on account of being useful, but not because it is innovative or interesting).

Reviewer #2:

Remarks to the Author:

The authors report an automated pipeline to register videos of brain activity measured with widefield calcium imaging to the Allen Institute's mouse brain atlas. The problem of cross-modality registration, in this case, activity to architecture, is important. The authors report two approaches: atlas-to-brain and brain-to-atlas. Having performant activity-to-architecture registration pipelines is valuable to neuroscience and data analysis communities. The authors have described their pipeline in detail and also provided useful video walk-throughs. Overall, I recommend publication after following improvements in a) comparing the pipeline with relevant literature and b) evaluating the accuracy of the registration.

1. Authors have missed an important brain-to-atlas registration pipeline published in Nature Communications (<https://www.nature.com/articles/ncomms11879>, <https://github.com/brainglobe/brainreg>). This pipeline may be useful for registration of calcium activity as well. The discussion should compare design choices between brainreg and mesonet pipelines.

2. Validating the accuracy of cross-modality registration methods is particularly challenging. I appreciated that the authors employed sensory inputs (a biological control) to validate their pipeline. However, it is not clear how plotting the DF/F signal quantifies the accuracy of registration. If the registration was erroneous by approximately half the size of the activated brain region, this analysis will not indicate. Authors need to augment their biological 'positive control' with image-based quantitative metrics. For example, stimuli are expected to activate a fairly large area of brain and their approximate sizes may be established. In that case, spatial correlation of DF/F signal around the identified landmarks is a useful metric of registration.

3. Likewise, it was difficult to assess the quality of registration with brain-to-atlas approach. The images in panel A are too small to see how raw, synthetic, and transformed data compare. Although clustering of activity motifs improves as registration progresses, there is no quantitative metric. The interpretation of quality depends on how many motifs were used for clustering and how the output is plotted. A metric such as Silhouette score is necessary. Silhouette score, for example, can clarify if the data suggests that 7 clusters provide the optimal clustering.

-- Shalin B. Mehta

Reviewer #3:

Remarks to the Author:

The paper presents an open-source suite to automatically estimate the location of brain areas from mesoscale imaging datasets. The authors combine two supervised learning methods to perform such alignment. The former is used to estimate 9 landmarks, and the latter to segment cortical boundaries. These features are then used to estimate affine transforms and warping to align each brain to a popular atlas. The authors use imaging of animals during sensory stimulation to show that the alignment consistently capture the relevant brain areas.

Major comments

Novelty. The papers combines a set of existing methods, and no new biological result is presented. Therefore, it seems that the originality is limited and the contributions mainly consists in the deployment of a software package that automates some of these steps.

Relevance. It has not been demonstrated by the authors that this precision is required when aligning the brains to an atlas. This might not be the case both because of the anatomical variability and because of the low resolution associated with this imaging technique. Most methods seem to rely on the alignment procedure as a rough initialization (see ref 10 for example, and works referred in it). Spatiotemporal factorization methods are a more reliable representation of functionally clustered areas. It is possible that there is an added advantage to it, but it should be demonstrated.

Claims. The authors claim this is a big data problem. However, in the reviewer's mind the current formulation only involves ~400 images, which do not seem to constitute such a computational challenge. There are also other unsubstantiated claims in the text (see minor comments).

Paper organization. The reviewer finds that the paper can be greatly improved in terms of clarity, precision, organization and writing. Concepts are scattered and not well explained in multiple points of the manuscript. English is often redundant and grammatically incorrect. Many sentences are difficult to parse.

Minor comments

58 — 61: not clear

77 — 83: not clear

86: not clear

90 — 91: not clear

96 — 97: not clear

Fig 2. A. It is probably not relevant depicting the whole neural network since it is exactly the one from DLC, not a contribution of the paper. A more parsimonious representation could work too. Same for Figure 3A.

Fig 2 B. The loss does not seem something needed in the main text, The reviewer suggests to move it to the supplement.

Fig 2 C. Colors are not explained, and crosses are used instead of filled circles in the "All" image. On my screen the colors of the brains look different. Not sure this is a local problem for me.

Have you tried Otsu's with in-painted plus closure?

"It is expected that the sensory stimulation paradigms activate the same analogous areas of the cortex across different mice while cortical mapping is relative ($\Delta F/F$) and not dependent on the basal level of fluorescence" -> not clear

Figure 5. These are all very qualitative arguments and images. It does not demonstrate that MesoNet achieves better results than other methods. Some more quantitative measure would help make the author's point better.

We found that the brain-to-atlas approach yielded fewer clusters (8 clusters (Fig. 6B(iii))) for transformed datasets than synthetic mis-aligned data (12 clusters (Fig. 6B(ii))), which misclassify some motif patterns as novel clusters because the spatial pattern is changed after rotation and resizing when quantified using the Phenograph classification method (Fig. 6B, 184 C).

—> The reviewer is not familiar with the clustering algorithm used, but the image (Fig 6Bii) seems to indicate that the number of clusters is smaller for the synthetic data. Perhaps printing the point cloud with an alpha value of about 0.2 might help better visualize the point the authors are trying to make?

241—247: what are the authors exactly implying with post-hoc fine tuning?

251-254: The reviewer considers that we really do not know how to modify nor what is the effect of modifying such hyper-parameters, besides trial and error of course.

256-257: in which point of the test this is demonstrated?

(LocaNMF) -> LocalNMF

285-186 -> this claim is not supported

The discussion seems long and repetitive

333-349: why different font?

English needs to be checked. Example: 430-432. But more issues in the text and above.

Reviewers' Comments:

Reviewer #1 (Expertise: Neuroimaging):

In this work Xiao and colleagues present a software package they have written to automatically delineate regions of dorsal cortex and align them to an atlas. I should preface my comments by stating that there is the possibility that I am quite fundamentally misunderstanding what the authors have done here (in which case I apologize, but would recommend to substantially rewrite the paper). I agreed to review the paper, having read the abstract, assuming the authors had developed a method to segment cortical areas based on calcium activity data (i.e. using the temporal activity patterns to delineate area boundaries). Having read the full paper, I am under the impression that they only use a single image per mouse and simply align this to an atlas (i.e. perform a slightly distorted affine transformation on a single image). The former would have been interesting and innovative. The latter is technically trivial and not worth writing a paper about. Having said that, I assume, if the software package is professionally written and supported, it will find use (on account of being useful, but not because it is innovative or interesting).

Response: initially, our intention was to align mesoscale GCaMP data based on both its functional activity pattern as well as brain and skull landmarks. Since the landmark based alignment procedure worked relatively well we had focused on that for the first submission of the paper. We do agree that there could be scenarios where landmarks were obscured, or where users would want to double-check data sets using a functional alignment. Here the spatial-temporal properties of the GCaMP signal would need to be taken into consideration. We now extend our MesoNet pipeline by adding three options of additional functional alignment steps.

First, we combined anatomical landmarks (predicted by DeepLabCut) with functional sensory maps (tail, visual, whisker stimulation) to align the reference atlas to brain image (sensory map based pipeline, Fig. 7a). Second, we developed a motif matching procedure to produce motif based functional maps (MBFMs) using the spatial-temporal activity patterns in calcium imaging data detected by seqNMF. The MBFMs were then used to train a U-Net model to directly predict anatomical atlas (MBFM-U-Net pipeline, Fig. 7b). Third, we take advantage of the MBFMs to train an unsupervised machine learning model VoxelMorph¹, to generate a deformation field for image registration (VoxelMorph pipeline, Fig. 7c). We are pleased to report that all of these three approaches achieve good performance for image alignment, and are now included as additional pipelines of MesoNet (Fig. 7, Supplementary Fig. 2, 4 and 5, and Supplementary Videos 1, 4, 5, and 6).

We now mention functional alignment in the abstract: “This anatomical alignment approach was extended by adding three functional alignment approaches that use sensory maps or spatial-temporal activity motifs.” (pg. 2 line 22-24).

We further discuss the functional alignment approaches. We have added to (pg. 9 line 175-230).

“Alignment using spatial-temporal functional cortical activity signals. An advantage of MesoNet is that most alignment can be performed using only a single raw fluorescent image (9 landmarks plus U-Net). MesoNet alignment is mostly dependent on cortical bone and brain edge markers and does not take into consideration internal functional boundaries. While this approach does show good correspondence with the location of expected sensory signals (Fig. 4), it would be advantageous to also make use of functional maps to reduce variability that exists between mouse lines with respect to neuro-anatomical and skull-based landmarks. Previously, we and others²⁻⁴ have used regional correlations of GCaMP signals during spontaneous activity to establish brain functional networks that correspond to underlying anatomical projections. While correlations yield robust maps, they do require placement of seed locations and some underlying assumptions of anatomical mapping⁴.

As a more reliable approach, we have employed a convolutional non-negative matrix factorization method (seqNMF)^{5,6} to recover stereotyped cortical activity motifs as a means of establishing functional maps. To perform seqNMF motif recovery an averaged mask (15 mice) was applied to limit the motif analysis from areas inside the brain window (brains roughly pre-aligned). One can also first do a brain-to-atlas alignment using MesoNet if the brains were significantly rotated or shifted between the experiments. As shown in Fig. 6, this approach can recover at least 6 major spatial-temporal activity motifs from each brain. This approach generates motif patterns that only require spontaneous activity and could therefore, be more advantageous than sensory modality mapping that requires specialized forms of stimulation (Fig. 7a). To create an aggregate picture of motif boundaries, we scaled each motif to its maximal value and then created a summed maximal intensity projection (Fig. 7b, c).

Like previous projections of seed pixel maps gradients⁴, projection of motifs led to the definition of functional “firewall” boundaries that reflect weighted activity transitions between major cortical groups of areas. Importantly, these firewalls were relatively stable across different animals (Supplementary Fig. 4) where functional resting state GCaMP activity was observed and can be used to create animal-specific motif based functional maps (MBFMs) (Fig. 7b, c). These MBFMs provide an opportunity to predict anatomical atlas (cortical overlay as the output of 9 landmarks plus U-Net in Fig. 4a) directly from another pre-trained U-Net model (Fig. 7b) that we call the MBFM-U-Net model.

MBFM can be combined with a deformable approach, such as VoxelMorph¹ (Fig. 7c, Supplementary Fig. 2e) to deform the template MBFM (Fig. 7d) so that its consensus internal structure would fit each mouse example. As the template MBFM is aligned with a reference atlas (Fig. 7d), so the deformation field predicted from VoxelMorph can be applied to the reference atlas to fit the functional regions in the input MBFM (atlas-to-brain). To check the performance of these mouse specific MBFM based alignments, we compared the predicted location of sensory regions for sensory map based (Fig. 7a), MBFM-U-Net (Fig. 7b), and VoxelMorph (Fig. 7c) pipelines. The accuracy of the prediction was then evaluated by measuring the Euclidean distance between the centroids of sensory stimulation induced activation and predicted atlas ROI centroids (Fig. 7e, f). All the three pipelines yielded similar distances to anatomical sensory map centres, although the VoxelMorph pipeline performed worse (in the barrel cortex BCS1 center, Fig. 7f).

The VoxelMorph pipeline's performance was improved by first applying a brain-to-atlas transformation to the MBFMs (Fig. 7f, BCS1). We further evaluated the performance of these pipelines by calculating the correlation coefficient between manually painted retrosplenial regions (RSP, ground truth, RSP consistently has clear boundaries in GCaMP functional data, Fig. 7d, e, Supplementary Fig. 1c) and predicted RSP regions. In this case, VoxelMorph performed significantly better than other pipelines as it was able to warp brain areas to fit functional boundaries in MBFMs (Fig. 7g, Supplementary Video 1). We suggest that, under certain conditions, there may be unique advantages to employing additional, computationally more-intensive steps such as VoxelMorph. These conditions might include analyses of specific lines of mice in which phenotypes affect neuroanatomical borders, or conditions such as lesions that may make alignment to the consensus atlas more challenging."

We have quantitatively evaluated the accuracy of these approaches and discussed the pros and cons. We have added to pg. 15 line 304-331:

"The VoxelMorph¹ pipeline offers an unsupervised machine learning method for the prediction of local deformation between pairs of images. We find that VoxelMorph provides a useful secondary alignment method - based on functional map features - which supplements our anatomical landmark-based alignment approach, improving its alignment with functional data. As such, we provide VoxelMorph as an optional add-in to the MesoNet pipeline.

For increased flexibility in registering different types of brain imaging data, MesoNet allows users to employ a landmark-based pipeline that relies on affine transformations to anatomical landmarks and/or functional sensory peak activations (Fig. 7a); a MBFM-U-Net-based pipeline that relies on a segmentation model predicting anatomical atlas from MBFMs (Fig. 7b); and a VoxelMorph-based pipeline that relies on model-based local deformation of the MBFM and the reference atlas (Fig. 7c). The anatomical landmark approach is most useful when landmarks are visible in the brain imaging data (our pipeline is flexible to employ a different number of landmarks to align the brain), as well as when brain images are rotated or shifted. However, its registration procedure is linear and therefore may be coarse when not used in combination with non-linear alignment methods. The sensory map based approach has the advantage of being able to utilize ground truth sensory induced activation areas, but this approach relies on experimental expertise and is highly susceptible to errors in the placement of the stimulation devices. The MBFM-U-Net approach is most useful for well-formed data with distinct functional features, but non-distinct anatomical landmarks. However, the model training of MBFM-U-Net is supervised and needs a well aligned label for each brain image, and its effectiveness may reduce if contrast and image features differ significantly from the training dataset. The VoxelMorph approach provides an unsupervised and deformable approach to align functional brain maps but it is less robust to larger rotations or shifts in cortical position in the frame (Supplementary Fig. 5). In order to address each dataset's individual strengths and weaknesses, MesoNet allows anatomical and functional approaches to be combined. A first brain-to-atlas alignment using a landmark-based pipeline combined with VoxelMorph could be a better option to align and deform an anatomical atlas to functional maps (Fig. 7g, Supplementary Video 1)."

Reviewer #2 (Expertise: Machine learning + (neuro-)imaging):

The authors report an automated pipeline to register videos of brain activity measured with widefield calcium imaging to the Allen Institute's mouse brain atlas. The problem of cross-modality registration, in this case, activity to architecture, is important. The authors report two approaches: atlas-to-brain and brain-to-atlas. Having performant activity-to-architecture registration pipelines is valuable to neuroscience and data analysis communities. The authors have described their pipeline in detail and also provided useful video walk-throughs. Overall, I recommend publication after following improvements in a) comparing the pipeline with relevant literature and b) evaluating the accuracy of the registration.

1. Authors have missed an important brain-to-atlas registration pipeline published in Nature Communications

(<https://www.nature.com/articles/ncomms11879>, <https://github.com/brainglobe/brainreg>). This pipeline may be useful for registration of calcium activity as well. The discussion should compare design choices between brainreg and mesonet pipelines.

Response: we appreciate the reviewers' suggestion of the brainreg software and the citation to the 2016 Nature Communications paper. Given the strong similarities in registration schemes, we have now included this paper in the discussion section. In order to fully evaluate the similarities and differences between MesoNet's and brainreg's approaches, we integrated the NiftyReg reg_aladin affine registration (the first and primary step of the brainreg pipeline, Niedworok et al., 2016) into our pipeline and evaluated the results. We found that NiftyReg - and, by extension, the brainreg pipeline - operates on and transforms 3D .nifti volumes as opposed to individual brain images, resulting in severe inconsistencies in extracting reasonably aligned data as output from this method. Furthermore, the unique vasculature of each mouse's brain that is present in calcium imaging data, combined with a lack of large-scale, clearly defined anatomical features beyond the stereotaxic landmarks that we already use, renders it difficult to register with brainreg's symmetric block-matching approach - which relies on locating corresponding points between an averaged template and any given image. Additionally, although brainreg further incorporates the reg_f3d (fast free-form deformation) registration method from NiftyReg as a further step, this method is based on similar feature-matching principles as the symmetric block-matching approach⁸. Lastly, recognizing the importance of local deformation techniques in improving image registration, we have integrated the VoxelMorph pipeline into MesoNet, including weights from a custom-trained VoxelMorph model that will be included with the software package. We conclude that brainreg is better suited to aligning images with highly distinct corresponding features - for example, histological sections of the brain, which have very distinct anatomical features and limited between-animal variance. In contrast, MesoNet is better suited to aligning images with limited anatomical commonalities and large differences in setup and image quality, such as functional calcium imaging. We further discuss the philosophy of this approach in the revised discussion section, as follows and repeated above (pg. 13 line 278-302):

“Other pipelines have been developed to facilitate automatic registration of various types of brain imaging data based on common features between images^{1,9}; however, these pipelines are primarily oriented towards aligning images with distinct anatomical features that remain relatively consistent across images. To evaluate the utility of these pipelines for functional

calcium images - which are often lacking in highly distinct anatomical features - we evaluated brainreg⁹ and VoxelMorph¹ pipelines as add-ins for MesoNet.

The brainreg pipeline offers a series of automated registration and segmentation steps suited to anatomical brain slice data⁹. As brainreg currently only operates on 3D volumes of anatomical brain images from specific online sources, we instead adapted its symmetric block-matching registration algorithm^{7,9} to our input data. This was accomplished by taking one 512x512 8-bit calcium image stack (repeated 10x) into a NIFTI volume after all MesoNet registration steps, and attempting to register it to a template averaged calcium image in the same fashion. This procedure, closely mirroring a key stage of the brainreg pipeline, did not produce consistent transformations - especially for images that were angled or lacking features that were present in the template image (e.g. the olfactory bulb). From these results, we determined that the feature-matching procedure utilized by brainreg requires a close correspondence in features between images. Such anatomical features are present and consistent across many types of histological brain slice data, but not in functional data such as that derived from mesoscale calcium images. Furthermore, vasculature - a highly visible feature across many calcium imaging datasets - is mostly unique across mice, providing further visual differentiation between images that challenges feature-matching approaches to image registration. As such, brainreg is primarily useful for registering anatomical images - such as histological brain slices - whereas MesoNet offers a novel approach to robust functional image registration, even for images with limited anatomical correspondence to a template.”

2. Validating the accuracy of cross-modality registration methods is particularly challenging. I appreciated that the authors employed sensory inputs (a biological control) to validate their pipeline. However, it is not clear how plotting the DF/F signal quantifies the accuracy of registration. If the registration was erroneous by approximately half the size of the activated brain region, this analysis will not indicate. Authors need to augment their biological 'positive control' with image-based quantitative metrics. For example, stimuli are expected to activate a fairly large area of brain and their approximate sizes may be established. In that case, spatial correlation of DF/F signal around the identified landmarks is a useful metric of registration.

Response: we agree with the reviewer’s comment regarding the DF/F signals with sensory mapping. We had included these signals to show the brains aligned using MesoNet could be used to develop temporal profiles for specific regions of Interest. We did not mean to imply that this was necessarily a way of checking alignment. In response to reviewer #1, we now include a scheme to make use of the peak activation of sensory maps as functional landmarks to further align the atlas (Fig. 7a). We choose to use the peak activation of sensory maps to avoid the variability of the shape and size of sensory map patterns in different mice caused by slightly different stimulation setups and individual responsiveness to sensory stimuli (Fig. 4c and Supplementary Fig. 3). We do confirm that the sensory map centers are in the expected locations based on the atlas (Fig. 4c, Fig. 7f, Supplementary Fig. 3 and Supplementary Fig. 4).

We have added to pg. 15 line 319-321: “The sensory map based approach has the advantage of being able to utilize ground truth sensory induced activation areas, but this approach relies on experimental expertise and is highly susceptible to errors in the placement of the stimulation devices.”

We also include a new alignment pipeline using spatial-temporal functional cortical activity signals that provide a means of functional activity registration. We now make it clear in the revised manuscript that any plots of DF/F are meant to show the utility of the system and not alignments.

We have added to the “**Alignment using spatial-temporal functional cortical activity signals.**” section (pg. 9 line 175-230).

3. Likewise, it was difficult to assess the quality of registration with brain-to-atlas approach. The images in panel A are too small to see how raw, synthetic, and transformed data compare. Although clustering of activity motifs improves as registration progresses, there is no quantitative metric. The interpretation of quality depends on how many motifs were used for clustering and how the output is plotted. A metric such as Silhouette score is necessary. Silhouette score, for example, can clarify if the data suggests that 7 clusters provide the optimal clustering.

-- Shalin B. Mehta

Response: we appreciate the reviewer mentioning the difficulty of assessing the brain-to-atlas approach. We have now quantified the performance of brain-to-atlas MesoNet by comparing MesoNet with manual labelled alignment. We are pleased to report that MesoNet shows high quality of brain-to-atlas alignment.

We have added to pg. 8 line 147-152: “**To further quantify the performance of brain-to-atlas alignment, we compared MesoNet with manual labelled alignment by calculating the Euclidean distance between the landmarks of the anterior tip of the interparietal bone and cross point between the median line and the line which connects the left and right frontal pole, and angle of the midline compared to the ground truth common atlas. MesoNet performs significantly better than manual labelled alignment in both comparisons (Fig. 5c, d).**”

We appreciate the reviewers’ suggestion of quantification of clustering. We have now done this and report silhouette scores for these figures (Fig. 6b). Importantly, the brain to atlas transformation leads to the best performance regarding clustering of activity motifs. We also provide high resolution images for all the figures.

We have added to pg. 9 line 170-173: “**We further quantified the clusters by calculating silhouette score, showing a better separation after brain-to-atlas transformation. The silhouette score calculated from raw data was 0.43, from mis-aligned synthetic data was 0.39 and the score from brain-to-atlas transformed data was the highest at 0.48, indicating clusters with the least overlap.**”

Reviewer #3 (Expertise: Neuroimaging):

The paper presents an open-source suite to automatically estimate the location of brain areas from mesoscale imaging datasets. The authors combine two supervised learning methods to perform such alignment. The former is used to estimate 9 landmarks, and the latter to segment

cortical boundaries. These features are then used to estimate affine transforms and warping to align each brain to a popular atlas. The authors use imaging of animals during sensory stimulation to show that the alignment consistently capture the relevant brain areas.

Major comments

Novelty. The papers combines a set of existing methods, and no new biological result is presented. Therefore, it seems that the originality is limited and the contributions mainly consists in the deployment of a software package that automates some of these steps.

Response: our goal was to present an open source alignment tool geared towards mesoscale functional images. We agree that our intention was not to present a series of novel experimental results but to have a robust tool. We have now extended the manuscript and its novelty based on the suggestion of yourself and reviewer 1. This extension includes using brain functional activity motifs for part of the alignment process. In this way, users would have the ability to perform a supplemental alignment based on functional data, or could do this if they did not have the structural landmarks for some reason. We now include this as a new figure (Figure 7) and an additional component of the software. This additional procedure was adapted from the VoxelMorph¹ pipeline, which we have integrated into MesoNet. VoxelMorph provides an unsupervised and deformable approach to align and deform an anatomical atlas to functional maps.

Another novelty of MesoNet is the automatic brain region delimitation using U-Net. We now further developed U-Net to directly predict anatomical atlas from functional motif data, not only delimitate the boundary of the brain from raw image.

We have added to pg. 23 line 500-521:

“Alternative pipelines. Depending on the format and contents of the data, one can select from different registration strategies to the one described above. If the brain images used do not have clearly defined anatomical landmarks (which are necessary for the landmark-based affine transformation), three strategies are available. First, the alignment approach can leverage cross-image commonalities in sensory-related activity. In our sensory map-based pipeline, MesoNet can detect four unilateral or bilateral activation peaks of calcium images and use these peaks as control points in a non-linear piecewise affine transformation implemented in scikit-image¹⁰ (PiecewiseAffineTransform). This additional step allows sensory stimulation activation (e.g. of tail, whiskers, and visual field; Fig. 4c) to be used for registration based on functional and not just anatomical information in the data (Fig. 7a).

Second, one can train a U-Net model on a set of motif-based functional maps (MBFMs, see “Generation of MBFM”) and label images (we used manually modified output masks from 9 landmarks plus U-Net MesoNet as label images, Fig. 4a), and then use this MBFM-U-Net model to directly predict the anatomical atlas on a new set of MBFMs. Third, one can train an unsupervised VoxelMorph model¹ on a set of MBFMs. After training, the VoxelMorph model can predict a deformation field between pairs of images. The deformation field can be used to transform template MBFM to each input MBFM. Specifically, using VoxelMorph terminology

the input MBFM is fixed, and the template MBFM is the moving image (Supplementary Fig. 2e, Supplementary Fig. 5). As the template MBFM is aligned with the reference atlas (Fig. 7d), we can apply the same deformation field to transform the reference atlas to fit the input MBFM. The deformation field is also exported for optional re-use in other analyses.”

Relevance. It has not been demonstrated by the authors that this precision is required when aligning the brains to an atlas. This might not be the case both because of the anatomical variability and because of the low resolution associated with this imaging technique. Most methods seem to rely on the alignment procedure as a rough initialization (see ref 10 for example, and works referred in it). Spatiotemporal factorization methods are a more reliable representation of functionally clustered areas. It is possible that there is an added advantage to it, but it should be demonstrated.

Response: we agree that wide field imaging does not require extremely high precision for alignment but there are reasons for doing this as a preliminary step. We also acknowledge that the 9 landmarks we have selected represent skull features and brain margins but not necessarily functional regions. We have evidence that cortical maps can also be precise at the mm scale and studies in the visual system report transitions occurring over 0.1 mm scales^{11,12}.

We have added to pg. 12 line 250-252: “The high precision of the image registration is required in some conditions such that cortical maps can be precise at the mm scale. Furthermore, studies in the visual system report transitions occurring over 0.1 mm scales^{11,12}.”

We agree that spatiotemporal factorization methods are a more reliable representation of functionally clustered areas. We now add a functional alignment method based on data derived from spatiotemporal factorization methods such as seqNMF.

We have added to pg. 9 line 187-189: “As a more reliable approach, we have employed a convolutional non-negative matrix factorization method (seqNMF)^{5,6} to recover stereotyped cortical activity motifs as a means of establishing functional maps.”

We also demonstrated that brain-to-atlas transformation leads to better performance regarding clustering of activity motifs in Fig. 6. In some cases, brain-to-atlas alignment may improve the performance of functional alignment pipelines. For example, we found rotated or shifted input images may reduce the performance of the VoxelMorph model (supplementary Fig. 5). On the other hand, applying a brain-to-atlas first to the input image may improve the performance of VoxelMorph (Fig. 7f, BCS1).

We have added to pg. 15 line 325-327: “The VoxelMorph provides an unsupervised and deformable approach to align functional brain maps but it is less robust to larger rotations or shifts in cortical position in the frame (Supplementary Fig. 5).”

And pg. 11 line 220-221: “The VoxelMorph pipeline’s performance was improved by first applying a brain-to-atlas transformation to the MBFMs (Fig. 7f, BCS1).”

Claims. The authors claim this is a big data problem. However, in the reviewer's mind the current formulation only involves ~400 images, which do not seem to constitute such a computational challenge. There are also other unsubstantiated claims in the text (see minor comments).

Response: we thank the reviewers for their suggestion. While ~400 images is not big data, manual alignment of the images will still require substantial labor and expertise. Fortunately, most alignment procedures in MesoNet can be made with only a single image (with the exception of functional map add-ons). We tested MesoNet on 400 brain images. It takes about 13 minutes to align all the brains, including automatic brain region delimitation and export .mat files of brain region ROIs for each image. This helps to analyse high throughput data generation platforms such as the recent automatic self-imaging home cage.

We have added to pg. 3 line 37-41: “Automatic registration and segmentation of brain imaging data can greatly improve the speed and precision of data analysis and does not require an expert anatomist. This is particularly crucial when using high-throughput neuroimaging approaches, such as automated mesoscale mouse imaging¹⁵, where the amount of data generated greatly exceeds the capacity of manual segmentation.”

We thank the reviewers for their suggestion, we agree that many speculative claims in the paper are not supported so we have now revised or removed them.

Paper organization. The reviewer finds that the paper can be greatly improved in terms of clarity, precision, organization and writing. Concepts are scattered and not well explained in multiple points of the manuscript. English is often redundant and grammatically incorrect. Many sentences are difficult to parse.

Response: we thank the reviewers for their suggestion and have revised and simplified the text to improve clarity and organization.

Minor comments

58 — 61: not clear

Response: we have revised this text.

We have added to pg. 4 line 54-56: “For the brain-to-atlas approach, our system automatically registers cortical images to a common coordinate framework using predicted cortical landmarks.”

77 — 83: not clear

Response: we have revised this text.

We have added to pg. 5 line 76-79: “While we see brain-to-atlas scaling as being the most appropriate method for aggregating experiments, MesoNet can handle special cases such as brains that have been imaged at different angles or brains that are partly out of frame, and will return a set of best fit regions of interest that can be matched with known anatomical regions by users.”

86: not clear

Response: we have revised and integrated this to the text above.

90 — 91: not clear

Response: we have revised this text.

We have added to pg. 5 line 81-83: “To transform a brain image to fit an atlas or rescale an atlas to the brain, the first step is to define the landmarks in a common coordinate system to align to the reference atlas.”

96 — 97: not clear

Response: we have revised and simplified this text.

We have added to pg. 5 line 86-89: “We then averaged brain images (images are manually aligned during experiments, $n = 12$ mice) to determine anatomical structures that fit a reference atlas (Supplementary Fig. 1b, c). We selected 9 clearly defined landmarks^{28,29} and created a common coordinate system while setting the skull landmark Bregma as (0,0 mm) (Fig. 1b, c; Table 1).”

Fig 2. A. It is probably not relevant depicting the whole neural network since it is exactly the one from DLC, not a contribution of the paper. A more parsimonious representation could work too. Same for Figure 3A.

Response: we thank the reviewers for their suggestion, we have now moved the figure of the DLC neural network from Fig 2a, and the figure of U-Net neural network from Fig 3a to Supplementary Fig.2.

Fig 2 B. The loss does not seem something needed in the main text, The reviewer suggests to move it to the supplement.

Response: we thank the reviewers for their suggestion, we have now moved Fig. 2b to Supplementary Fig.2.

Fig 2 C. Colors are not explained, and crosses are used instead of filled circles in the “All”

image. On my screen the colors of the brains look different. Not sure this is a local problem for me.

Response: we thank the reviewers for their suggestion, we now make sure that colours are clearly indicated and change the colours of crosses in Fig 2a.

Have you tried Otsu's with in-painted plus closure?

Response: we thank the reviewers for their suggestion. In our understanding, "in-painted plus closure" means using an in-painting method to close the output mask to remove artifacts, such as blood vessels. We tried this method and did not get better results than U-Net.

"It is expected that the sensory stimulation paradigms activate the same analogous areas of the cortex across different mice while cortical mapping is relative ($\Delta F/F$) and not dependent on the basal level of fluorescence" -> not clear

Response: we thank the reviewers for their suggestion and have now clarified the text in the paper.

We have added to pg. 7 line 125-127: "Cortical mapping is presented in terms of relative activation ($\Delta F/F$) and is not strictly dependent on the basal level of GCaMP calcium-induced fluorescence."

Figure 5. These are all very qualitative arguments and images. It does not demonstrate that MesoNet achieves better results than other methods. Some more quantitative measure would help make th author's point better.

Response: we thank the reviewers for their suggestion; we have now added a quantitative measurement and comparison of brain-to-atlas alignment in Fig 5c, d.

We have added to pg. 8 line 147-152: "To further quantify the performance of brain-to-atlas alignment, we compared MesoNet with manual labelled alignment by calculating the Euclidean distance between the landmarks of the anterior tip of the interparietal bone and cross point between the median line and the line which connects the left and right frontal pole, and angle of the midline compared to the ground truth common atlas. MesoNet performs significantly better than manual labelled alignment in both comparisons (Fig. 5c, d)."

We found that the brain-to-atlas approach yielded fewer clusters (8 clusters (Fig. 6B(iii))) for transformed datasets than synthetic mis-aligned data (12 clusters (Fig. 6B(ii)), which misclassify some motif patterns as novel clusters because the spatial pattern is changed after rotation and resizing when quantified using the Phenograph classification method (Fig. 6B, 184 C). —> The reviewer is not familiar with the clustering algorithm used, but the image (Fig 6Bii) seems to indicate that the number of clusters is smaller for the synthetic data. Perhaps printing the point cloud with an alpha value of about 0.2 might help better visualize the point the authors are trying to make?

Response: we thank the reviewers for their suggestion; we have now changed the alpha value to 0.2. We further quantified the clusters by calculating the silhouette score, showing after brain-to-atlas transformation the clusters are better separated.

We have added to pg. 9 line 170-173: “We further quantified the clusters by calculating silhouette score, showing a better separation after brain-to-atlas transformation. The silhouette score calculated from raw data was 0.43; the score from mis-aligned synthetic data was 0.39; and the score from brain-to-atlas transformed data was the highest at 0.48, indicating a cluster with the least overlap.”

241—247: what are the authors exactly implying with post-hoc fine tuning?

Response: we thank the reviewers for their suggestion. We have now removed this text.

251-254: The reviewer consider that we really do not know how to modify nor what is the effect of modifying such hyper-parameters, besides trial and error of course.

Response: we thank the reviewers for their suggestion. We have now removed this text.

256-257: in which point of the test this is demonstrated?

Response: we thank the reviewers for their suggestion. We agree that this claim is not tested so we have now removed it.

(LocaNMF) -> LocalNMF

Response: LocaNMF is the name of the method introduced by Saxena et al. (2020).

285-186 -> this claim is not supported

Response: we thank the reviewers for their suggestion, we agree that this speculative claim is not supported so we have now removed it.

The discussion seems long and repetitive

Response: we have streamlined and organized the discussion section to flow more logically and avoid repetition.

333-349: why different font?

Response: we have converted all text in the document to Times New Roman.

English needs to be checked. Example: 430-432. But more issues in the text and above.

Response: we have clarified the language in pg. 21 line 454-457: “The three points that are used

are the ones that the model most accurately predicted (based on the output of the sigmoid activation function by TensorFlow via DeepLabCut²²). If specific points were not chosen as part of the analysis, the first three points selected in each hemisphere are used for the transformation.” as well as throughout the paper.

References

1. Balakrishnan, G., Zhao, A., Sabuncu, M. R., Gutttag, J. & Dalca, A. V. VoxelMorph: A Learning Framework for Deformable Medical Image Registration. *IEEE Trans. Med. Imaging* (2019) doi:10.1109/TMI.2019.2897538.
2. White, B. R. *et al.* Imaging of functional connectivity in the mouse brain. *PLoS One* **6**, e16322 (2011).
3. Mohajerani, M. H. *et al.* Spontaneous cortical activity alternates between motifs defined by regional axonal projections. *Nat. Neurosci.* **16**, 1426–1435 (2013).
4. Vanni, M. P., Chan, A. W., Balbi, M., Silasi, G. & Murphy, T. H. Mesoscale Mapping of Mouse Cortex Reveals Frequency-Dependent Cycling between Distinct Macroscale Functional Modules. *J. Neurosci.* **37**, 7513–7533 (2017).
5. Mackevicius, E. L. *et al.* Unsupervised discovery of temporal sequences in high-dimensional datasets, with applications to neuroscience. *Elife* **8**, (2019).
6. MacDowell, C. J. & Buschman, T. J. Low-Dimensional Spatiotemporal Dynamics Underlie Cortex-wide Neural Activity. *Curr. Biol.* **30**, 2665–2680.e8 (2020).
7. Modat, M. *et al.* Global image registration using a symmetric block-matching approach. *J Med Imaging (Bellingham)* **1**, 024003 (2014).
8. Modat, M. *et al.* Fast free-form deformation using graphics processing units. *Comput. Methods Programs Biomed.* **98**, 278–284 (2010).
9. Niedworok, C. J. *et al.* aMAP is a validated pipeline for registration and segmentation of high-resolution mouse brain data. *Nat. Commun.* **7**, 11879 (2016).
10. van der Walt, S. *et al.* scikit-image: image processing in Python. *PeerJ* **2**, e453 (2014).

11. Garrett, M. E., Nauhaus, I., Marshel, J. H. & Callaway, E. M. Topography and areal organization of mouse visual cortex. *J. Neurosci.* **34**, 12587–12600 (2014).
12. Zhuang, J. *et al.* An extended retinotopic map of mouse cortex. *Elife* **6**, (2017).
13. Murphy, T. H. *et al.* High-throughput automated home-cage mesoscopic functional imaging of mouse cortex. *Nat. Commun.* **7**, 11611 (2016).

Reviewers' Comments:

Reviewer #2:

Remarks to the Author:

The authors now compare their pipeline with the brainReg, and point out that their pipeline works better for registering activity to atlas. The revision addresses this and other concerns I raised. I suggest publication.

Reviewer #3:

Remarks to the Author:

The authors addressed in details many of my comments and I praise the introduction of the activity-map based alignment. The novelty of the manuscript is still limited. However, the proposed set of approaches, if well organized and documented in a single software package, will represent a useful tool for the community. I would recommend publishing this article if the points below are meticulously addressed.

1) I have found difficult to parse some portions of the manuscript. In particular, the new sections of the paper describing the functional alignment procedures. For instance, the authors write:

56 "We extended this anatomical
57 alignment approach with three options of functional alignment steps making use of functional
58 sensory maps and spontaneous cortical activity motifs. The cortical activity motifs were used to
59 generate functional maps (MBFMs) for the training of an unsupervised VoxelMorph model or
60 another U-Net model (MBFM-U-Net) to directly predict the anatomical atlas from the spatial
61 structure of functional activity."

- The authors introduce the word "functional map" without a citation or explanation. And the abbreviation MBFMs does not match what comes before the parenthesis!
- The authors introduce VoxelMorph without any explanation (it is explained in details the discussion, not ideal), the sentences should be self-contained and explanatory. A reference is not enough in the reviewers mind. What is an unsupervised VoxelMorph model? The authors should introduce the intuition.
- to predict the anatomical atlas -> what does this mean? Predict a deformation, a location?

The reviewer suggests, when writing a paragraph or section, to start with the intuition, and go into the details, and then discuss the exceptions/alternatives. It is important to make sure the concepts are introduced in the right order. Mixing all this up makes it very difficult to follow. (Example lines 188-198).

2) The reviewer was not able to understand whether one needs to align brains with landmarks before aligning with the functional approaches? Maybe adding a flow diagram/cartoon explaining all the possible combinations of processing steps and the order in which they should be carried out might help.

Minor comment

244-245: functional repeated twice

We outline reviewer/editor queries in blue, our response in black, and copied elements of the revised manuscript in red text below.

EDITOR'S/REVIEWER'S SUMMARY COMMENTS

You will see that the other two reviewers find that your revisions improved the manuscript, but a few important points remain to be addressed. Seeing as R1 had previously questioned the extent of the conceptual novelty, and this was reiterated by R2 in this round, we have agreed that the implementation of the analysis methods in an accessible and well-documented end-to-end package is an invaluable aspect of the work for Nature Communications. For that reason, we ask that you take the time during this revision to address R2's remaining comments, but also to thoroughly test and document your code so it can be an accessible tool for the community. This will -hopefully- maximize its utility and facilitate its adoption by a wide number of users later on.

Response: We appreciate the reviewers' emphasis on implementing an accessible and well-documented end-to-end package. To this end, we have packaged MesoNet, accompanying pre-trained models, sample data, and Matlab code for motif-based functional map (MBFM) into thoroughly documented Code Ocean capsules (Python: [10.24433/CO.1919930.v1](https://codeocean.com/capsules/10.24433/CO.1919930.v1) and Matlab: [10.24433/CO.4985659.v1](https://codeocean.com/capsules/10.24433/CO.4985659.v1)) that allow users to observe the code's dependencies and operation and reproduce the paper results. Secondly, we added and thoroughly tested six end-to-end automated pipelines to enable users to quickly output results from input images (**Supplementary Fig. 6 and Supplementary Video 10**). We offered both an easy-to-use GUI and a powerful command-line interface (CLI), allowing users to integrate the toolbox with their own workflow. We also provide a step-by-step, customizable way to use MesoNet with an annotated Google Colab(https://colab.research.google.com/github/bf777/MesoNet/blob/master/mesonet_demo_colab.ipynb) that allows users to run MesoNet in a cloud environment using our sample data or their own data, enabling our package to be used without any installation or specialized computing resources on the part of the user. Additionally, we have thoroughly documented our code, both in docstrings embedded within functions as well as through an illustrated GitHub wiki (<https://github.com/bf777/MesoNet/wiki>) accompanying our code's repository, the contents of which are also available in .pdf and .txt form in our main OSF repository at <https://osf.io/svztu/>.

Reviewer #2 (Remarks to the Author):

The authors now compare their pipeline with the brainReg, and point out that their pipeline works better for registering activity to atlas. The revision addresses this and other concerns I raised. I suggest publication.

Response: We thank the reviewer for their support of our publication as well as for their feedback and suggestions, which helped enhance the depth of our publication.

Reviewer #3 (Remarks to the Author):

The authors addressed in details many of my comments and I praise the introduction of the activity-map based alignment. The novelty of the manuscript is still limited. However, the proposed set of approaches, if well organized and documented in a single software package, will represent a useful tool for the community. I would recommend publishing this article if the points below are meticulously addressed.

Response: We appreciate the reviewer's praise of our activity-map based alignment. We now provide additional figures, raw data repositories, example videos, and code that provide an end-to-end, well-documented solution for mesoscale image alignment (see description of repositories above in EDITOR's SUMMARY COMMENTS). We anticipate that these features will make Mesonet both a novel and valuable research tool.

Discussion of novelty. The most common method to segment the cortex is based on an anatomic reference atlas¹. The advantage of this approach is the consistency across different studies and research groups, making it convenient to compare results from various studies. This consistency relies on the accurate annotation of the landmarks and overlay of the reference atlas, which requires substantial labor and expertise. We apply machine learning models to automate the registration and overlay of the reference atlas and the segmentation of brain regions on mesoscale wide-field images with high accuracy. However, anatomical reference atlases may fail to track the dynamic organization of functional cortical modules in different sensory and cognitive processes, as well as the precise topography of brain parcellation. An alternative approach is to define cortical regions based on activity and generate unique atlas for individual animals. Previous methods include grouping pixels using clustering analyses^{2,3} and extracting functional modules using non-negative matrix factorization⁴ or independent component analysis⁵. Compared with anatomical atlases, cortical segmentation derived from neural activity can more faithfully represent the functional organization of the cortex in individual animals. They may also detect neural dynamics localized in regions that do not correspond to standard areas in anatomic atlases. However, functional modules often vary across individual animals and different studies^{2,3,5} using different methods. Different research groups also use varied terminology to refer to regions in their functional atlases. All of these factors make it difficult to compare and interpret results across studies. To address this, we developed novel animal-specific motif-based functional maps (MBFMs) that represent cortical consensus patterns of regional activation that can be used for brain registration and segmentation. Furthermore, our automated pipelines can be combined to consider both anatomical consistency and individual functional variations to help better analyze regional brain activity data. Our open-source platform, MesoNet, allows researchers to register their functional maps to a common atlas framework based on cortical landmarks and will help comparisons across studies.

We have added and repeated above (pg. 15 line 325 - 332): “Overall, we apply machine learning models to automate the registration and overlay of the reference atlas and the segmentation of brain regions using mesoscale wide-field images with high accuracy. We developed novel animal-specific motif-based functional maps that represent cortical consensus patterns of regional activation that can be used for brain registration and segmentation. Our automated pipelines can be combined to consider both anatomical consistency and functional individual variations to help better analyze brain regional activity. Our open-source platform, MesoNet, allows researchers to register their functional maps to a common atlas framework based on cortical landmarks and will help comparisons across studies.”

1) I have found difficult to parse some portions of the manuscript. In particular, the new sections of the paper describing the functional alignment procedures. For instance, the authors write:

56 "We extended this anatomical
57 alignment approach with three options of functional alignment steps making use of functional
58 sensory maps and spontaneous cortical activity motifs. The cortical activity motifs were used
to
59 generate functional maps (MBFMs) for the training of an unsupervised VoxelMorph model or
60 another U-Net model (MBFM-U-Net) to directly predict the anatomical atlas from the spatial
61 structure of functional activity."

- The authors introduce the word "functional map" without a citation or explanation. And the abbreviation MBFMs does not match what comes before the parenthesis!

Response: We now define a functional map more clearly as a repeated spatial-temporal activation domain observed with a functional imaging indicator. Functional indicators can include but are not limited to genetically encoded calcium sensors.

We have revised and clarified the language in pg. 4 line 56-68: “This alignment approach, while robust in the presence of anatomical landmarks, does not leverage regional patterns within functional calcium imaging data that are related to underlying structural connectivity^{2,30,31}. We suggest that functional maps that represent specific spatio-temporal consensus patterns of

regional activation observed using activity sensors such as GCAMP6^{2,30,31} or potentially hemodynamic activation^{32,33} can also be used for atlas registration. As such, we extended this anatomical alignment approach with three pipelines that can use functional sensory maps or spontaneous cortical activity. Spontaneous cortical activity was assessed by recovering regional activity motifs³⁴ and using them to generate motif-based functional maps (MBFMs). MBFMs can then be used to train a learning-based framework, VoxelMorph³⁵, which nonlinearly deforms the reference atlas to register it to the brain image. A MBFM based U-Net model (MBFM-U-Net) can directly predict positions of anatomical brain regions from the spatial structure of MBFMs.”

- The authors introduce VoxelMorph without any explanation (it is explained in details the discussion, not ideal), the sentences should be self-contained and explanatory. A reference is not enough in the reviewers mind. What is an unsupervised VoxelMorph model? The authors should introduce the intuition.

Response: We thank the reviewers for their suggestions. We have now added context to introduce the intuition behind using VoxelMorph; we have also clarified the statement “unsupervised VoxelMorph model” and changed it to “learning-based framework.” Unlike traditional registration methods that optimize an objective function for each pair of images, which can be time-consuming for large datasets or rich deformation models, VoxelMorph builds on recent learning-based methods for fast, deformable, pairwise medical image registration. We say “unsupervised” because the model training of VoxelMorph not requiring any label data for the training set.

We have added to pg. 10 line 211-215: “To supplement our anatomical landmark-based alignment approach, we capture local deformation using functional map features by integrating VoxelMorph³⁵ as an optional add-in to the MesoNet pipeline (Fig. 7c, Supplementary Fig. 2e). VoxelMorph offers a learning-based approach that determines a deformation field that is required for the transformation and registration of image pairs such as MBFMs.”

- to predict the anatomical atlas -> what does this mean? Predict a deformation, a location?

Response: We thank the reviewer for their comment. We have revised and clarified the language in pg. 10 line 206-209: “These MBFMs provide an opportunity to predict brain regional boundaries (represented by a cortical overlay as the output of 9 landmarks plus U-Net in Fig. 4a) using another pre-trained MBFM based U-Net model (Fig. 7b) that we call the MBFM-U-Net model.”

The reviewer suggests, when writing a paragraph or section, to start with the intuition, and go into the details, and then discuss the exceptions/alternatives. It is important to make sure the concepts are introduced in the right order. Mixing all this up makes it very difficult to follow. (Example lines 188-198).

Response: We thank the reviewer for their suggestion and have revised and simplified the text to improve clarity and organization. We have clarified this section as the reviewer suggests, by starting with the intuition, the details, and then discuss the exceptions/alternatives.

We have added to pg. 10 line 190-194: “As a potentially less-biased approach, we employ seqNMF^{34,46} (as in Fig. 6) to recover stereotyped cortical spatio-temporal activity motifs as a means of establishing functional maps. This approach generates motif patterns that only require spontaneous activity and would be advantageous over sensory modality mapping that requires specialized forms of stimulation and additional imaging trials (Fig. 4, Fig. 7a).”

2) The reviewer was not able to understand whether one needs to align brains with landmarks before aligning with the functional approaches? Maybe adding a flow diagram/cartoon explaining all the possible combinations of processing steps and the order in which they should be carried out might help.

Response: We thank the reviewer for their comment and suggestion. We now include **Supplementary Fig. 6** outlining basic and combination pipelines to enable users to apply MesoNet to their data efficiently. For the VoxelMorph pipeline, we suggest aligning brains with landmarks before aligning with the motif-based functional maps (**Supplementary Fig. 6f**). We also discuss the advantages of the various methods and define scenarios where investigators would want to employ specific pipelines.

We have revised and clarified the language in pg. 15 line 317-323: “The VoxelMorph approach provides a fast learning-based (unsupervised) framework for deformable registration, but it is less robust to larger rotations or shifts in cortical position in the frame (Supplementary Fig. 5). In order to address each dataset’s individual strengths and weaknesses, MesoNet allows anatomical and functional approaches to be combined. We suggest first applying brain-to-atlas alignment using a landmark-based pipeline then combining this with VoxelMorph as a better option to align and deform a reference atlas to functional maps (Supplementary Fig. 6f).”

Minor comment

244-245: functional repeated twice

Response: We thank the reviewer for their comment and have removed the redundant second mention of “functional” before “brain images.”

We have revised and clarified the language in pg. 12 line 252-253: “In order to carry out image registration, we leveraged multiple sources of anatomical and functional landmarks derived from brain images as well as skull junctions.”

References:

1. Wang, Q. *et al.* The Allen Mouse Brain Common Coordinate Framework: A 3D Reference Atlas. *Cell* **181**, 936–953.e20 (2020).
2. Barson, D. *et al.* Simultaneous mesoscopic and two-photon imaging of neuronal activity in cortical circuits. *Nat. Methods* **17**, 107–113 (2020).
3. Lake, E. M. R. *et al.* Simultaneous cortex-wide fluorescence Ca²⁺ imaging and whole-brain fMRI. *Nature Methods* vol. 17 1262–1271 (2020).
4. Saxena, S. *et al.* Localized semi-nonnegative matrix factorization (LocaNMF) of

widefield calcium imaging data. *PLoS Comput. Biol.* **16**, e1007791 (2020).

5. Makino, H. *et al.* Transformation of Cortex-wide Emergent Properties during Motor Learning. *Neuron* **94**, 880–890.e8 (2017).